1    FINAL MANUSCRIPT

3    MS No.: os-2019-11

# ESTIMATION OF PHYTOPLANKTON PIGMENTS FROM OCEAN-COLOR SATELLITE OBSERVATIONS IN THE SENEGALO-MAURITANIAN REGION BY USING AN ADVANCED NEURAL CLASSIFIER

By

Khalil Yala[1], N'Dye Niang[2], Julien Brajard[1,4], Carlos Mejia[1], Maurice Ouattara[2], Roy El Hourany[1], Michel Crépon[1] and Sylvie Thiria[1,3]

**ESTIMATION OF PHYTOPLANKTON PIGMENTS FROM OCEAN-COLOR**
**SATELLITE OBSERVATIONS IN THE SENEGALO-MAURITANIAN REGION BY**
**USING AN ADVANCED NEURAL CLASSIFIER**
By
Khalil Yala[1], N'Dye Niang[2], Julien Brajard[1,4], Carlos Mejia[1], Maurice Ouattara[2], Roy El
Hourany[1], Michel Crépon[1] and Sylvie Thiria[1,3]
[1] IPSL/LOCEAN, Sorbonne Université (Université Paris6, CNRS, IRD, MNHN), 4 Place
Jussieu, 75005 Paris, France
[2] CEDRIC, CNAM, 292 rue Saint Martin, 75003 Paris, France
[3] UVSQ, F-78035, Versailles, France
[4] Nansen Center, Thormøhlensgate 47, 5006, Bergen, Norway
Corresponding author: Michel Crepon (crepon@locean-ipsl.upmc.fr)
**ABSTRACT**
We processed daily ocean-color satellite observations to construct a monthly climatology of
phytoplankton pigment concentrations in the Senegalo-Mauritanian region. Our proposed new method
primarily consists of associating, in well-identified clusters, similar pixels in terms of ocean-color
parameters and in situ pigment concentrations taken from a global ocean database. The association is
carried out using a new Self Organized Map (2S-SOM). Its major advantage is to allow taking into
account the specificity of the optical properties of the water by adding specific weights to the different
ocean color parameters and the in situ measurements. In the retrieval phase, the pigment concentration
of a pixel is estimated by taking the pigment concentration values associated with the 2S-SOM cluster
presenting the ocean-color satellite spectral measurements, which are the closest to those of the pixel
under study according to some distance. The method was validated by using a cross-validation
procedure. We focused our study on the fucoxanthin concentration, which is related to the abundance
of diatoms. We showed that the fucoxanthin starts to develop in December, presents its maximum
intensity in March when the upwelling intensity is maximum, extends up to the coast of Guinea in
April and begins to decrease in May. The results are in agreement with previous observations and
recent in situ measurements. The method is very general and can be applied in every oceanic region.

## 1 - INTRODUCTION

Phytoplankton are the basis of the ocean food web and consequently drive the ocean productivity.
They also play a fundamental role in climate regulation by trapping atmospheric carbon dioxide ($CO_2$)
through gas exchanges at the sea surface, and consequently lowering the rate of anthropogenic increase
in the atmosphere of $CO_2$ concentration by about 25% (*Le Quéré et al, 2018*). With the growing interest
in climate change, one may ask how the different phytoplankton populations will respond to changes
in ocean characteristics (temperature, salinity, acidity) and nutrient supply, which presents an
important societal impact with respect to both climate and fisheries, with a possible effect on fish
grazing phytoplankton via the marine food chain.
Methods for identifying phytoplankton have greatly progressed during the last two decades.
Phytoplankton were first described by microscopy. Microscopy is time consuming and is unable to
identify picoplankton. Imaging flow cytometry (IFC) has renewed microscopic methods, thanks to the
speed at which they are able to characterize phytoplankton in a water sample (IOCCG report n°15,
2014). An alternative method is the analysis of seawater samples by high-performance liquid
chromatography (HPLC) which is widely used to categorize broad phytoplankton groups such as PFT
or PSC (*Jeffreys et al*, 1997, *Brewin et al,* 2010, *Hirata et al,* 2011). HPLC enables the identification
of 25 to 50 pigments within a single analysis, which is much easier and faster to conduct than
microscopic observations (*Sosik, H.M et al,* 2014*). Each phytoplankton group is associated with
specific diagnostic pigments, and a conversion formula, the so-called "Diagnostic Pigment Analysis"
can be derived to estimate the percentage of each group from the pigment measurements (*Vidussi et
al,* 2001; *Uitz et al*, 2010). HPLC measurements are now recognized as the standard for calibrating
and validating satellite-derived chlorophyll-a concentration and for mapping groups of phytoplankton
(IOCCG report n°15, 2014).
The use of satellite ocean color sensor measurements has permitted to map the ocean surface at a daily
frequency. Satellite sensors measure the sunlight, at several wavelengths, backscattered by the ocean.
The downwelling sunlight interacts with the seawater through backscattering and absorption in such a
manner that the upwelling radiation transmitted to the satellite ('water-leaving' reflectance) contains
information related to the composition of the seawater. The light transmitted to the satellite depends
on the phytoplankton cell shape (backscattering), its pigments (absorption), the dissolved matter (e.g.
CDOM).
This upwelling radiation, the so-called remotely sensed reflectance $\rho_w(\lambda)$, is determined by the spectral
absorption $a$ and backscattering ($b_b$ ($m^{-1}$)) coefficients of the ocean (pure water and various particulate
and dissolved matters) using the simplified formulation (*Morel* and *Gentili*, 1996):

$\rho_w(\lambda) = G\, b_b\,(\lambda)/(a(\lambda) + b_b(\lambda))$        (1)


where ($a$ (m$^{-1}$) ) is the sum of the individual absorption coefficients of water, phytoplankton pigments,
colored dissolved organic matter, and detrital particles, ($b_b$ (m$^{-1}$) ) depends on the shape of the
phytoplankton species. $G$ is a parameter mainly related to the geometry of the situation (sensor and
solar angles) but also to environmental parameters (wind, aerosols).
In the open ocean far from the coast (in case-1 waters), the light seen by the satellite sensor mainly
contains information on phytoplankton abundance and diversity. Ocean-color measurements have
been first used intensively to estimate chlorophyll-*a* concentration (*chl-a* in the following) in the
surface waters of the ocean, marginal seas and lakes. (*Longhurst et al.,* 1995; *Antoine et al.,* 1996;
*Behrenfeld and Falkowski*, 1997; *Behrenfeld et al.,* 2005; *Westberry et al.*, 2008).
It has been shown that it is also possible to extract additional information such as phytoplankton size-
classes (PSC) by using some relationship between chlorophyll concentration and PSC (*Uitz et al.*, 2006;
*Ciotti and Bricaud*, 2006; *Hirata et al.*, 2008; *Mow and Yoder,* 2010). These algorithms try to establish
a relationship between the *chl-a* concentration and the *chl-a* concentration fractions associated with
each of the three PSC. Some of them (*Uitz et al*, 2006; *Aiken et al.,* 2009) break-down the *chl-a*
abundance into several ranges for each of which a specific relationship is computed. Others (*Brewin*
*et al*, 2010; *Hirata et al*, 2011) are based on a continuum of *chl-a* abundance. Studies have also been
done to estimate the phytoplankton groups (PFT) by taking into account spectral information
(*Sathyendranath et al.,* 2004, *Alvain et al.*, 2005, 2012; *Hirata et al.*, 2011; *Ben Mustapha et al,* 2013;
*Farikou et al*, 2015). This is of fundamental interest to the understanding of the phytoplankton behavior
and to modeling its evolution.
Due to highly non-linear relationship linking the multispectral ocean color measurements with the
pigment concentrations, we proposed a neural network clustering algorithm (2S-SOM) able to deal
with multi variables linked by complex relationships. The 2S-SOM algorithm is well adapted to this
complex task by weighting the different inputs. The clustering algorithm was calibrated on a restricted
database composed of remote sensed observations co-located with measurements taken in the global
ocean.
In the present paper, we propose the retrieval of the major pigment concentrations from satellite ocean
color multi-spectral sensors in the Senegalo-Mauritanian upwelling, which is an oceanic region off the
coast    of    West    Africa    where    a    strong    seasonal    upwelling    occurs    (Figure    1).

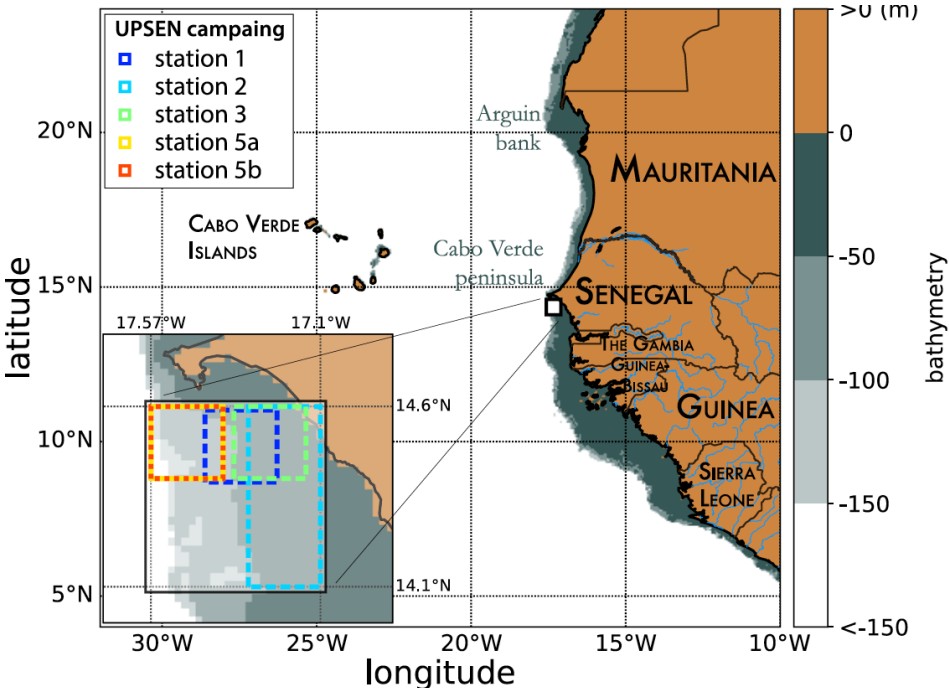

Figure 1: *Mauritania and Senegal coastal topography. The land is in brown and the ocean depth is represented in meters by the color scale on the right side of the figure. The UPSEN stations are shown at the bottom left cartoon of the figure.*

The Senegalo-Mauritanian upwelling is one of the most productive eastern boundary upwelling systems (EBUS) with strong economic impacts on fisheries in Senegal and Mauritania. Since the region has been poorly surveyed in situ, we have chosen to extract pertinent biological information from ocean-color satellite measurements. The region has been intensively studied by analysis of SeaWiFS ocean-color data and AVHRR sea-surface temperature as reported in *Demarcq* and *Faure* (2002), *Sawadogo et al.* (2009), *Farikou et al.* (2013, 2015), *Ndoye et al,* (2014) and more recently by *Capet et al,* (2017) with in situ observations.

The paper is articulated as follows: in section 2, we present the data we used (in situ and remote sensing observations). The mathematical aspect of the clustering method (2S-SOM) is detailed in section 3. In section 4 we present the methodological results. The spatio-temporal variability of the fucoxanthin and chl-a concentration in the Senegalo-Mauritanian upwelling region are presented in section 5, as well as the results of the oceanic UPSEN campaigns. In section 6 we discuss the results and the method. A conclusion is presented in section 7.

**2- MATERIALS**

In this study we used three distinct datasets: the first was used to calibrate the method, the second to
conduct a climatological analysis of the Senegalo-Mauritanian upwelling region and the third was
obtained during the oceanographic UPSEN campaign. These datasets are composed of satellite remote
sensing observations and in-situ measurements.

*2.1 The calibration data base (DPIG)*
The calibration database (DPIG) comprises in situ pigment measurements co-located with satellite
ocean-color observations done by the SeaWiFS (Sea-viewing, Wide-Field-of-view Sensor).
This DPIG is composed of 515 matched satellite observations and in situ measurements made in the
global ocean (mainly in the North Atlantic and the equatorial ocean; *Ben Mustapha et al.*, 2014). The
match-up criteria were quite severe: we used satellite pixel situated at a distance less than 20km from
the in situ measurement in a time window of +/- 12h. The geographic distribution of the 515 coincident
in situ and satellite measurements is shown in Fig. 2. Matchup procedure between in situ and satellite
observation is a crucial question to estimate remote sensing algorithms. If the parameters of the
procedure are too severe, the number of collocated data is

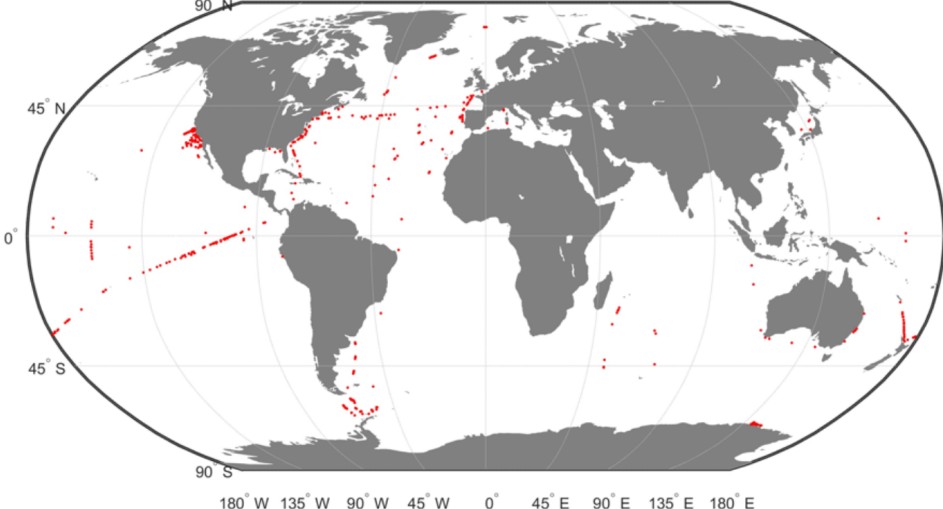



Figure 2: *Geographic positions of the 515 in situ and satellite collocated measurements of the*
*DPIG database.*

dramatically decreasing. If the parameters are too large, it is the accuracy of the matching, which is
decreasing. We accordingly chose some compromise. Usually people use a matchup window of 3X3 pixels
(*Alvain et al,* 2005) which corresponds to a distance less than 20km between the satellite pixel and in
situ measurement, since we deal with level 3 satellite observations whose pixel is of the order of 9X9km.
This criterium refers to the typical length of ocean variability (*Levy et al*, 2012; *Levy*, 2003)

In Figure 3 we present the $R^2$ coefficient between the in situ *chl-a* a and the SeaWiFS *chl-a* a computed
by using the OC4V4 algorithm (*O'Reilly et al,* 2001) for the DPIG collocated observations. We remark
that the two measurements are in good agreement at global scale. Each data of DPIG is a vector


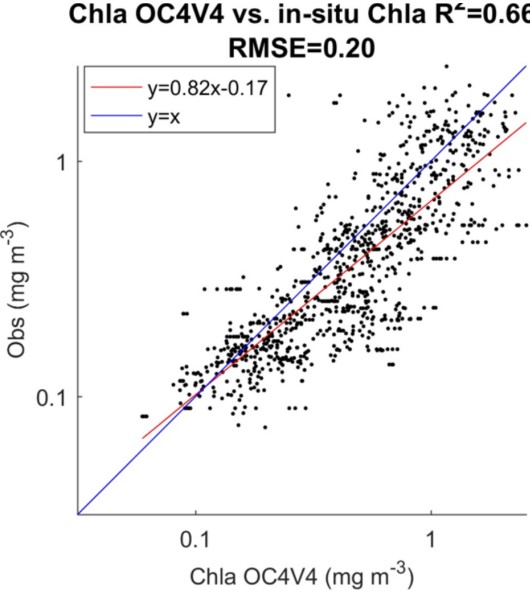



Figure 3: *Dispersion diagram of DPIG chl-a computed from the SeaWiFS observations using the*
*OC4V4 algorithm versus in situ chl-a. The coefficient of vraisemblance $R^2$ and the RMSE (Root Mean*
*Square Error) were computed in mg m$^{-3}$*

having 17 components (five ocean reflectance ($\rho_w(\lambda)$ and *Ra($\lambda$)* at five wavelengths (412, 443, 490,
510 and 555nm)*,* SeaWiFS *chl-a,* five in situ pigment ratios and in situ *chl-a* concentration). The in
situ *chl-a* a concentration ranges between 0.007 and 3. mg m$^{-3}$ (see Table 1).
The five *Ra($\lambda$)* are defined following *Alvain et al,* (2012 :

*Ra($\lambda$) = $\rho_W(\lambda)$/ $\rho_{Wref}$($\lambda$, chl-a)*              (2)

where the parameter $\rho_{wref}(\lambda, chl_a)$ is an average reflectance depending on the *chl-a* concentration only
which was computed according to the procedure reported in *Farikou et al*, 2015. *Ra($\lambda$)* is a non-
dimensional parameter which depends on the *chl-a* abundance at second order and is mainly sensitive
to the secondary pigments (*Alvain et al* , 2012).

The DPIG database thus provides information on the existing links between the pigment composition
and the SeaWiFS measurements. The pigment composition are defined by the pigment ratios which
are non-dimensional variables of the form in the present study:

Pigment Ratio=DP/T*chl-a*                                    (3)

which is defined as the ratio of the diagnostic pigment (DP) versus the total *chl-a*
(T*chl-a* = *chl-a* +divinyl *chl-a*, according to *Alvain et al.*, 2005).

The pigments of the DPIG and their statistical characteristics are given in Table 1. The statistical tests
presented in Figure 3 ($R^2$ and RMSE) and in Table 1 (MEAN, STD, MIN, MAX) were computed in
mg m$^{-3}$.


|  | RDIVINY A | RPERID | RFUCO | R19HF | RZEAX | CHLORO *IN SITU* |
|---|---|---|---|---|---|---|
| MEAN | 0.1414 | 0.0272 | 0.1248 | 0.1859 | 0.1696 | 0.5292 |
| STD | 0.1584 | 0.0196 | 0.0971 | 0.0996 | 0.2063 | 0.5720 |
| MIN | 0.0037 | 0.0035 | 0.0053 | 0.0066 | 0.0027 | 0.007 |
| MAX | 0.8889 | 0.2027 | 0.8514 | 0.7654 | 1.5574 | 2.9980 |



Table 1: *Pigments of the DPIG and their statistical characteristics: STD (Standard Deviation), MIN*
*(minimum value), MAX (maximum value).*

***2.2 The Senegalo-Mauritanian upwelling satellite data (DSAT)***
The satellite dataset we processed to retrieve the pigment concentration consist of five $\rho_w(\lambda)$ and five
*Ra(λ)* at five wavelengths (412, 443, 490, 510 and 555nm), and the SeaWiFS *chl-a* concentration
observed in the Senegalo-Mauritanian upwelling region (8°N-24°N, 14°W-20°W; Figure 3) during 11
years (1998-2009) by SeaWiFS. This data set is here below denoted *DSAT*.
The satellite observations ($\rho_w(\lambda)$ and *chl-a* concentration) were provided by NASA with a resolution
of nine kilometers. Due to the presence of Saharan dusts in this region, very few estimations of satellite
$\rho_w(\lambda)$ and in situ *chl-a* were available, and some satellite estimations of *chl-a* could present strong over-
estimations (*Gregg et al*, 2004). For this reason, we reprocessed the $\rho_w(\lambda)$ and *chl-a* data with an
atmospheric correction algorithm developed specifically for Saharan dust (*Diouf et al,* 2013,
http://poac.locean-ipsl.upmc.fr) in order to improve the satellite observations.

### *2.3 The UPSEN database*

Recently, some HPLC measurements were made in the Senegalo-Mauritanian region during two
oceanographic cruises (UPSEN campaigns) of the oceanographic ship "Le Suroit" from 7 to 17 March
2012 and from 5 to 26 February 2013 as reported in *Ndoye et al*, (2014); *Capet et al*, (2017). The goal
was to study the dynamics and the biological variability of the Senegalo-Mauritanian upwelling.
During these campaigns, in-situ HPLC measurements were carried out. We expected to be able to co-
locate them with the ocean-color VIIRS (Visible Infra-red Imaging Radiometer Suite) sensor
observations whose wavelengths are close to those of the SeaWiFS. Unfortunately, we were only able
to process satellite observations made on 21 February 2013 due to the presence of clouds and Saharan
aerosols the other days. We processed the satellite observations provided by the VIIRS sensor at four
wavelengths (443, 490, 510, 555 nm) for pixels in the vicinity of the ship stations (within a distance
of 20km) and observed in a time window of +/- 12h, and for which the satellite *chl-a* was less than
3 mg m$^{-3}$, which is the limit of validity of our method imposed by the range of *chl-a* observed in DGIP
(mean of 0.52 mg m$^{-3}$). Only five stations off Cabo Verde peninsula fitted these requirements (see
Figure 1 for their positions).

### 3 - THE PROPOSED METHOD (2S-SOM)

Classification methods were applied for retrieving geophysical parameters from large databases in
several studies including weather forecasting (*Lorenz*, 1969; *Kruizinga and Murphy*, 1983), short-term
climate prediction (*Van den Dool*, 1994), downscaling (*Zorita and von Storch*, 1999), reconstruction
of oceanic pCO$_2$ (*Friedrichs and Oschlies.*, 2009), and of *chl-a* concentration under clouds (*Jouini et*
*al*, 2013). In the present study, we used a new neural network classifier, which is an extension of the
SOM algorithms.

### *3-1 The SOM clustering*

The SOM algorithms (*Kohonen,* 2001) constitute powerful nonlinear unsupervised classification
methods. They are unsupervised neural classifiers, which have been commonly used to solve
environmental problems (*Cavazos,* 1999; *Hewitson et al,* 2002; *Richardson et al,* 2003; *Liu et al,* 2005,
2006; *Niang et al,* 2003, 2006; *Reusch et al,* 2007). The SOM aims at clustering vectors $z_i \in \mathbb{R}^N$ of a
multidimensional database $D$. Clusters are represented by a fixed network of neurons (the SOM map),
each neuron $c$ being associated with the so-called referent vector $w_c \in \mathbb{R}^N$ representing a cluster. The
self-organizing maps are defined as an undirected graph, usually a rectangular grid of size *p x q*. This
graph structure is used to define a discrete distance (denoted by $\delta$) between two neurons of the $p$ x $q$
rectangular grid which presents the shortest path between two neurons. Each vector $z_i$ of $\boldsymbol{D}$ is assigned
to the neuron whose referent $\boldsymbol{w}_c$ is the closest, in the sense of the Euclidean distance: $\boldsymbol{w}_c$ is called the
projection of the vector $z_i$ on the map. A fundamental property of a SOM is the topological ordering
provided at the end of the clustering phase: close neurons on the map represent data that are close in
the data space. The estimation of the referent vectors $\boldsymbol{w}_c$ of a SOM and the topological order is achieved
through a minimization process in which the referent vectors $\boldsymbol{w}$ are estimated from a learning data set
(The DPIG data base in the present case). The cost function is shown in Annex:
The SOMs have frequently been used in the context of completing missing data (*Jouini et al*, 2013),
so the projected vectors $z_i$ may have missing components. Under these conditions, the distance between
a vector $z_i \in \boldsymbol{D}$ and the referent vectors $\boldsymbol{w}_c$ of the map is the Euclidean distance that considers only the
existing components (the Truncated Distance or *TD* hereafter).

### *3-2 The 2S-SOM Classifier*
In the present case, we used the 2S-SOM algorithm, which is a modified version of the SOM*,* very
powerful in the case of a large number of variables. It automatically structures the variables having
some common characters into conceptually meaningful and homogeneous blocks. The 2S-SOM takes
advantage of this structuration of $\boldsymbol{D}$ and the variables into different blocks, which permits an automatic
weighting of the influence of each block and consequently of each variable. The block weighting
facilitates the clustering procedure by considering the most pertinent variables. The vectors of DPIG
defined in section 2 can be decomposed in four blocks. The essence of this decomposition in blocks is
that each of the 17 components of the DPIG vectors gathered information with a different physical
influence in the classification phase. The composition of each block is done as follows:
***First Block*** (B1) comprises the five pigment in-situ concentration ratios (divinyl chlorophyll-a,
peridinin, fucoxanthin, 19'hexanoyloxyfucoxanthin, zeaxanthin concentration ratios). The pigment
ratios are defined in Eq. 3.
***Second Block*** (B2) comprises the water-leaving reflectance $\rho_w(\lambda)$ at the five SeaWiFS wavelengths
***Third Block*** (B3) comprises the five $Ra(\lambda)$ ,
***Fourth Block*** (B4) comprises two variables: The in situ and the SeaWiFS *chl-a* concentrations.

The 2S-SOM is able to deal with a large quantity of variables, choosing those that are the most
significant for the classification and neutralizing those which are the least significant. This is done by
estimating weights on the blocks and the variables. We fully describe the 2S-SOM algorithm in Annex.
In the following we use a simplified version of 2S-SOM in which only the blocks are weighted.

*3.3 The calibration phase*
Similarly to the standard SOM, the 2S-SOM is determined through a learning phase by using a more
complex cost function (see Annex) that estimate for each neuron, in addition to the referent vector, a
weight ($\alpha$) for each block. For a neuron $c$, we define the weights $\alpha_{cb}$ of each block $b$ ($b = 1....4$). .
At the end of the calibration phase, each element $z_i$ of the dataset DPIG is associated with a referent
$w_c$ whose components are partitioned into four blocks. In the present study, the 2S-SOM map is
represented by a two-dimensional (9x18=162) grid that represents the partition of the DPIG dataset
into different classes. Each class provided by the 2S-SOM is associated with a so-called referent vector
$w_c$ with $c \in \{1.....162\}$. The size of the map has been determined by using the procedure provided by
the SOM software available at : http://www.cis.hut.fi/projects/somtoolbox/download/.

*3.4 The Pigment retrieval*
In the second phase, which is an operating phase, we estimated the pigment concentration ratios of a
pixel $PX_m$ from its satellite ocean-color sensor observations only. The 11 ocean color satellite
observations (5 $\rho_w(\lambda)$, 5 $Ra(\lambda)$, and *chl-a* ) of pixel $PX_m$ were projected onto the 2S-SOM using the
Truncated Euclidian Distance (section 3.1). We select the neuron $c$ associated with a referent vector
whose the 11 ocean-color parameters are the closest to those observed by the satellite sensor. The
pigment ratios of $PX_m$ are those associated with the neuron $c$. At the end of the assignment phase, each
pixel $PX_m$ of a satellite image is associated with a referent vector $w_c$, which has 6 pigment
concentration ratios among its 17 components. The flowcharts of the method (2S-SOM learning and
pigment retrieval) are presented in Figure 4.


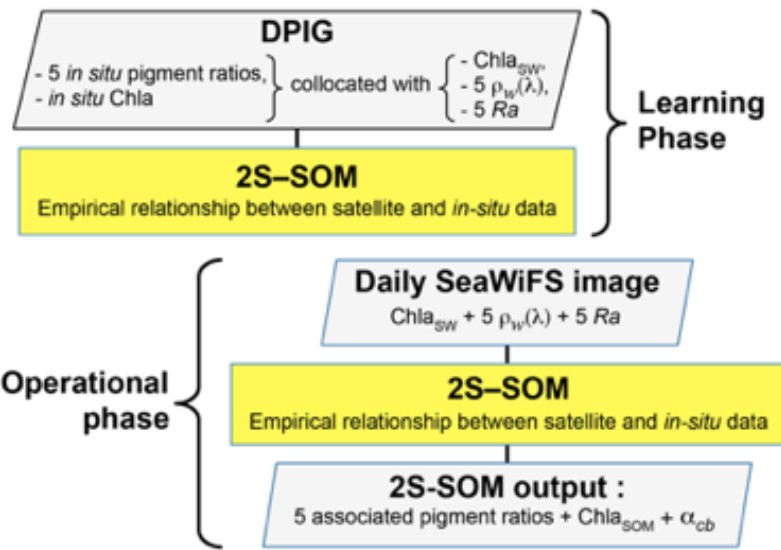

Figure 4: *Flowchart of the method: top panel - Learning phase; bottom panel – operational phase which consists in pigment retrieval and the determination of the $\alpha_{cb}$ block parameters.*

## 4 - METHODOLOGICAL RESULTS

### 4-1 Statistical validation of the method

The validation of the method was focused on the retrieval of the fucoxanthin ratio, which is a characteristic of diatoms, but the same procedure could be applied to any pigment. The hyper-parameter $\mu$ (see Annex) was optimized in order to retrieve that ratio, while $\eta$ was set constant since only the block were weighted in the present study. Due to the small amount of data in the DPIG, we estimated the accuracy of the fucoxanthin retrieval by a cross-validation procedure, which is a powerful procedure in statistics. The principle is the following: we learned 30 2S-SOM using 30 different learning datasets $L_i$ constituted of 90% of DPIG taken at random, and then computed statistical estimator on the retrieved quantities using 30 test datasets (10% of DPIG). The algorithm was as follows:

$i$=1 …. 30

1. determination at random of a learning dataset $L_i$ (90% of DPIG) and a test dataset $TL_i$ (10% of DPIG)

2. training of a 2S-SOM map $M_i$ using $L_i$ (see section 3.2 and 3.3).

3. Validation using $TL_i$ according to the procedure described in section 3.4

4 Estimation of the $RMSE_i$ and $R^2_i$ on $TL_i$ between the estimated and observed fucoxanthin ratios

*end*

Computation of the mean RMSE and $R^2$   $(R^2, \text{RMSE} = \frac{1}{30} \sum_{i=1}^{I=30} R^2 i, RMSEi)$

The flowchart of the cross-validation procedure is presented in Figure 5.

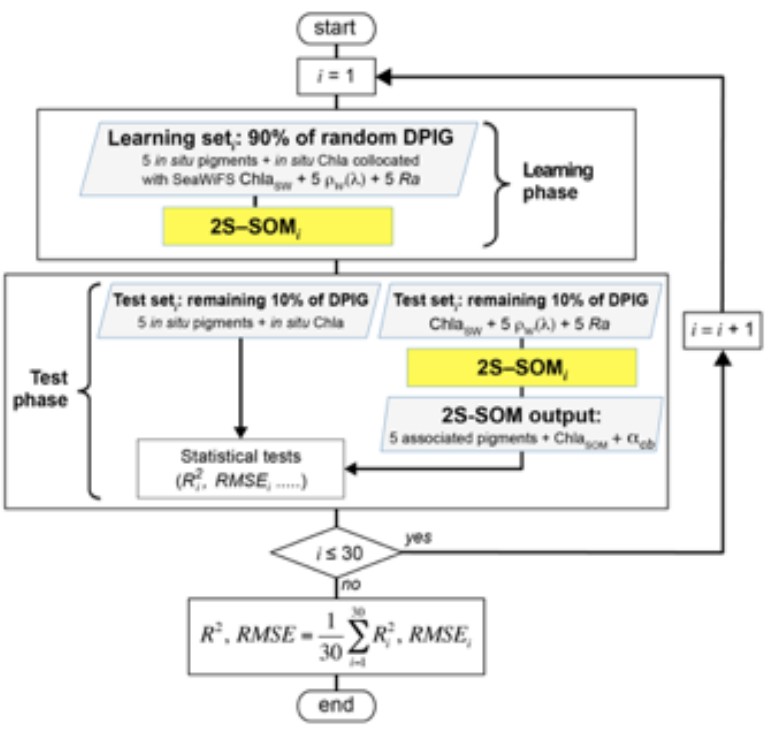



Figure 5: *Flowchart of the cross-validation procedure for 30 partitions of the DPIG database.*

Statistical parameters ($R^2$ coefficients, RMSE and P-values) of the cross validation between the DPIG
in situ pigments and the pigments given by the 2S-SOM averaged for the 30 2S-SOM realizations,
which are presented in table 2, show the good performance of the method.


|  | $R^2$ | RMSE (MG M$^{-3}$) | PVAL |
|---|---|---|---|
| CHLA SOM | 0.84 | 0.22 | 0.001 |
| DVCHLA | 0.60 | 0.02 | 0.001 |
| FUCO | 0.87 | 0.02 | 0.001 |
| PERID | 0.81 | 0.01 | 0.001 |



Table 2: *Statistical parameters ($R^2$ coefficients, RMSE and P-values) of the cross validation between*
*the DPIG in situ pigments and the pigments given by the 2S-SOM averaged for the 30 2S-SOM*
*realizations.*


### 4-2 Analysis of the topology of the 2S-SOM

As explained in sections 3-2 and 3-3, the referent vector components ($w_c \in R^{17}$), which are estimated
during the learning phase, are partitioned in four blocks B1, B2, B3 and B4. The hyper parameters $\mu$
was tuned in order to favor the accuracy of the retrieval of the fucoxanthin ratio. We recall that all the
pigment ratios are estimated during the calibration phase, but in the present paper attention was focused
on the fucoxanthin ratio when selecting the parameter $\mu$. In Figure 6, we

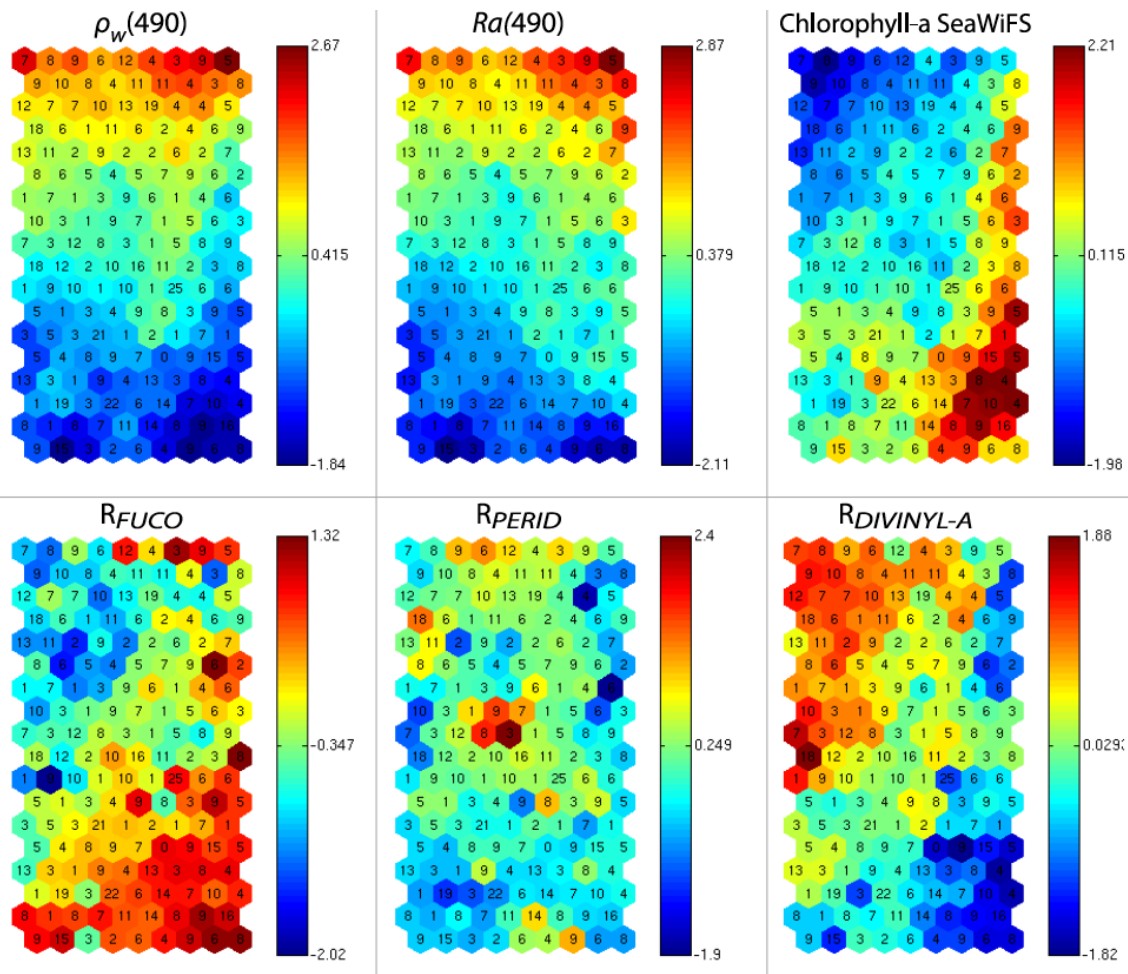


Figure 6: *2S-SOM Map. From left to right and top to bottom, values of the referent vectors for $\rho_w(490)$,*
*Ra(490), SeaWiFS chl-a, and fucoxanthin, peridinin, divinyl Ratios. The number in each neuron indicates the*
*amount of DPIG data captured at the end of the learning phase, the values indicated by the color bars are*
*centered-reduced and non-dimensional values.*

present six of the referent vector components of the 2S-SOM map. These components are $\rho_w(490)$,
*Ra(490)*, SeaWiFS *chl-a*, and the ratios of fucoxanthin, which is a specific diatom pigment, and of
*peridinin* and *divinyl*. They exhibit a coherent topological order, the components having close values
being close together on the topological map. The remaining eleven components (not shown) exhibit
the same coherent topological order. One can observe a very good topological order for the fucoxanthin
ratio that was favored by the determination of the hyperparameter $\mu$. Moreover, the bottom right region
in the 2S-SOM map (Figure 6) may correspond to the diatoms with a good confidence since high
fucoxanthin is associated with high chlorophyll concentration and low peridinin. This is endorsed in
section 5 by looking at the geographical location of the different pigment concentrations (figures 8, 10,
11). Another important remark is that the value of each component presents a large range of variation

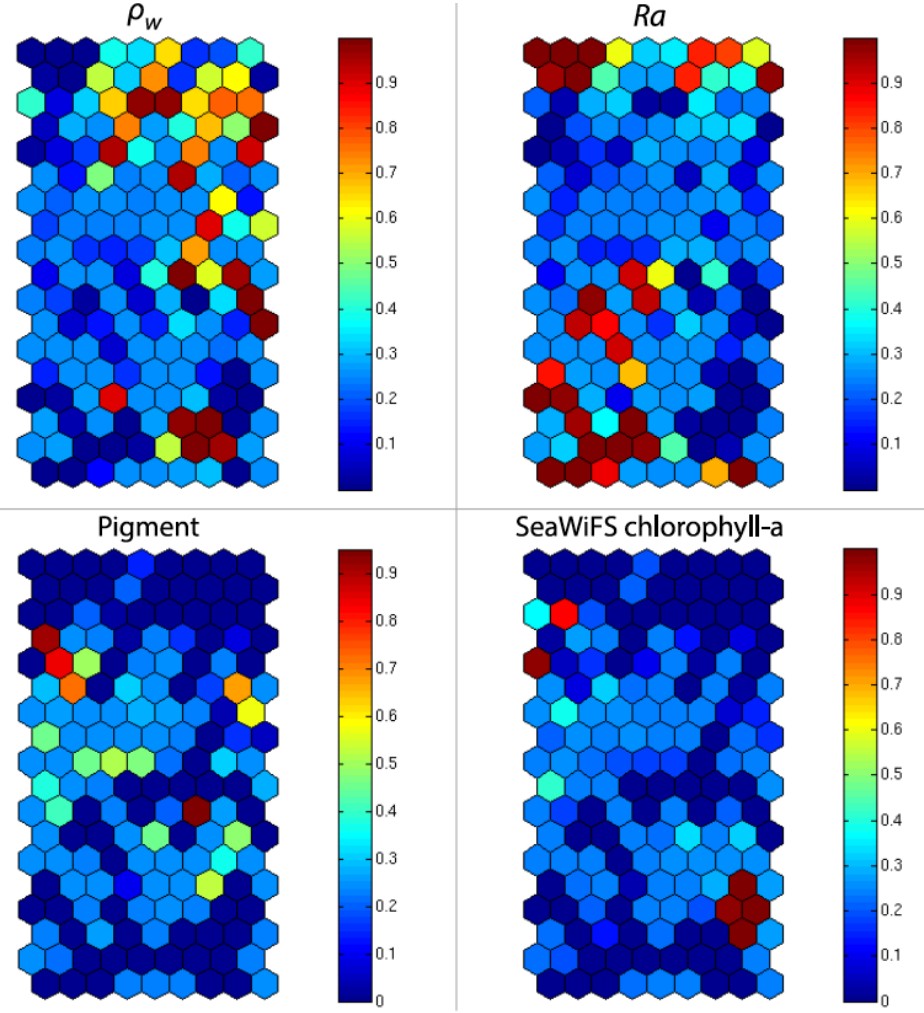



Figure 7: *2S-SOM map. Weights ($\alpha_{cb}$) of the four block parameters determined at the end of the learning*
*phase; from left to right and top to bottom: $\rho_w$, Ra, Pigment, SeaWifs chl-a. The color bars show the % of*
*the weight estimated by 2S-SOM, a value of 1 or 0 indicating that the data in the neuron are assembled with*
*respect to that block only.*

of the same order as the range of variation found in the DPIG variables. It means that the 2S-SOM
map has captured most of the variability of the dataset.
Figure 6 shows a strong link between the values of the referent vectors for fucoxanthin and *chl-a* (high
fucoxanthin and *chl-a* values, at the bottom right of the 2S-SOM) while fucoxanthin is high and *chl-a*
low for the referent vectors at the bottom left of the 2S-SOM. Additional information will be provided
by the *Ra(490)* values when the fucoxanthin is less closely linked to the chlorophyll.
Besides, for each neuron, the 2S-SOM provides a weight for each block ($\alpha_{cb}$) and each variable ($\beta_{cbj}$).
For a given neuron $c$ the weights ($\alpha_{cb}$) of the blocks are normalized, their sum being 1. A value of 1
for one block (and therefore a value of 0 for the other blocks) indicates that the data in the neuron are
gathered with respect to that block only because there is too much noise in the variables in the other
blocks. By examining the weights on the map, one can see which block most influences the link
between the satellite measurements and the pigment ratios.
In Figure 7, we present the $\alpha_{cb}$ values estimated during the learning phase of the 4 blocks (B1, B2, B3,
B4). For some neurons, only the blocks related to the reflectance and the reflectance ratio are used for
the definition of the neuron, while the weights for the two other blocks (pigments and *chl-a*) are null,
indicating that for these neurons, in situ observations and SeaWiFS *chl-a* are more noisy than the
reflectance. These neurons correspond to very small *chl-a* concentrations, which are estimated with
large error. Besides, we remark that high $\alpha$ values for *chl-a* corresponds to high *chl-a* concentration
values (bottom right of the *chl-a* panel in figure 7 and figure 6 respectively). For these cases, the
clustering assembled data that mainly depend on *chl-a* concentration.


**5 - GEOPHYSICAL RESULT**

In the present study, we apply the 2S-SOM (section 3), which explicitly makes a weighted use of the
data according to their specificity (ocean-color signals or in situ observations) to retrieve the
fucoxanthin concentration from remote sensed data in the Senegalo-Mauritanian upwelling region
where in situ measurements are lacking. According to the good results of the cross-validation method
as shown in section 4.1, we expect that the 2S-SOM will provide pertinent results in a region which
has been poorly surveyed.


*5-1 The pigment estimation from SeaWiFS observations in the Sénégalo-Mauritanian upwelling region*

We decoded the DSAT database (section 2-3) using the 2S-SOM for 11 years (1998-2009) of SeaWiFS data observed in the Senegalo-Mauritanian upwelling region (8°N-24°N, 14°W-20°W). This study was done according to the retrieval phase described in section 3.4. For each day, we projected the 11 SeaWiFS observations (5 $\rho_w(\lambda)$, 5 $Ra(\lambda)$ and *chl-a*) of each pixel $PX_m$ on the 2S-SOM. At the end of the assignment phase, each pixel of a satellite image was associated with 6 pigment concentration ratios. The underlying assumption is that the link between the remote sensing information and the pigment ratios of a pixel is this provided by the selected referent $w_c$. Thanks to the topological order provided by the 2S-SOM, we expect that the best neurons chosen during the retrieval would give accurate concentration ratios. In Figures 8, 10 and 11 we present the fucoxanthin concentration ratio restitution for three different days and the associated SeaWiFS Chlorophyll images (1 and 6 January, and 28 February 2003). Due to the limited size of the DPIG, the range of the ratio learned for the

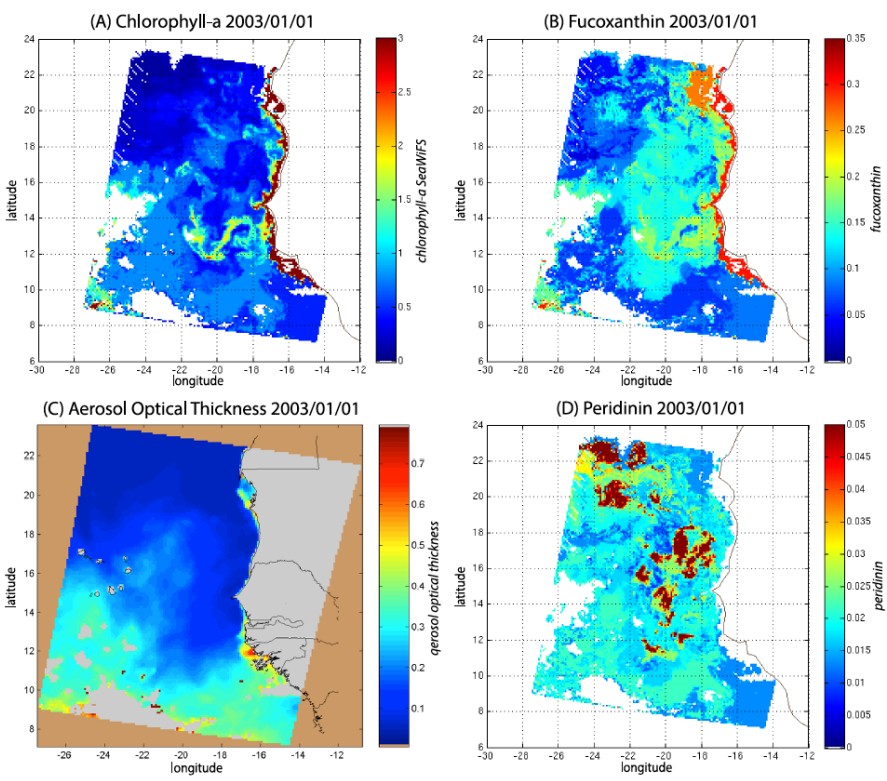

Figure 8: *A) chl-a concentration, (B) fucoxanthin ratio, (C) aerosol optical thickness, (D) peridinin for 1 January 2003. Panels (B) and (D) show that a second-order information was retrieved, which is correlated with the chl-a concentration (A) but not equivalent. The aerosol optical thickness (C) does not seem to contaminate the estimated parameters (fucoxanthin and peridinin ratios).*

the fucoxanthin is between 0.3% and 20% with a mean of 10% and the *chl-a* content is between 0.5
mg m$^{-3}$ and 3 mg m$^{-3}$. The statistical estimator we used cannot extrapolate what has not been learned,

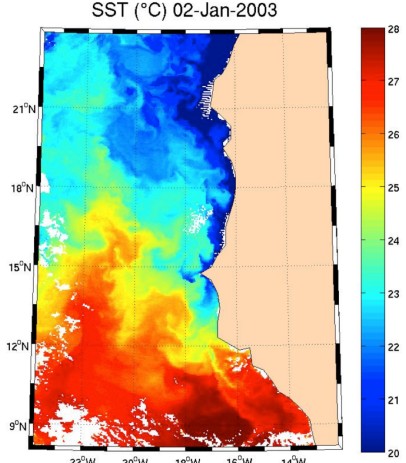


Figure 9: *SST for 2 January 2003. Note the well-marked upwelling (cold temperature) north of 13°N.*

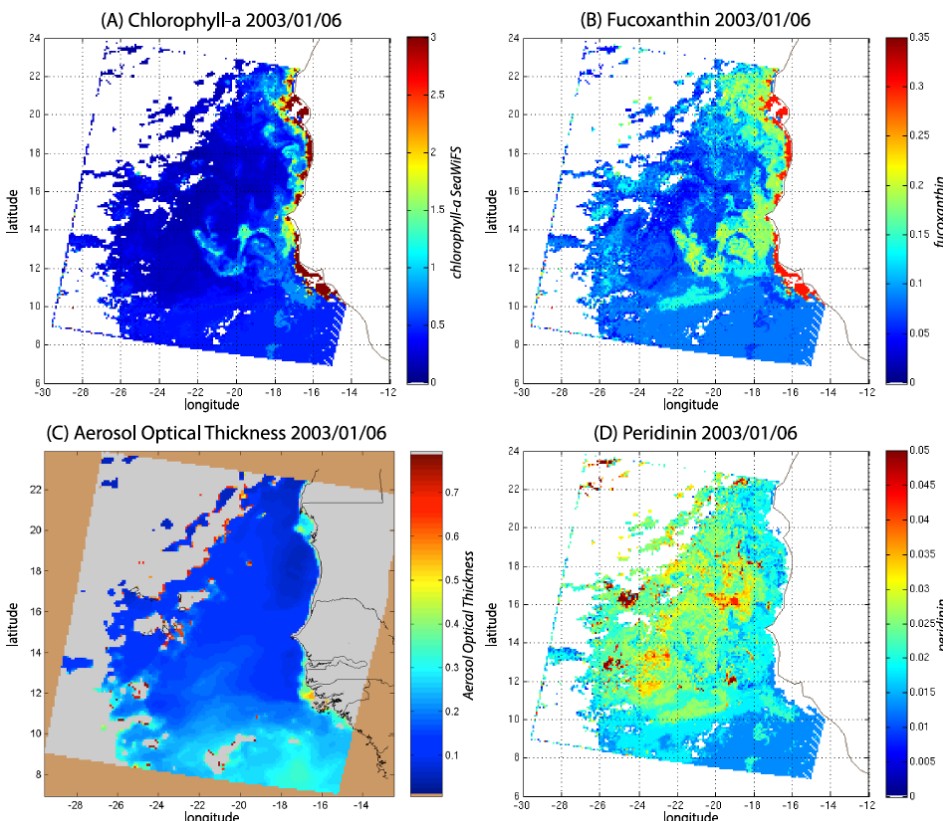


Figure 10: *(A) chl-a concentration, (B) fucoxanthin ratio, (C) aerosol optical thickness, (D) peridinin for 6*
*January 2003. Panels (B) and (D) show that a second-order information was retrieved, which is correlated*
*with the chl-a concentration (A) but is not equivalent. It is found that the aerosol optical thickness (C) does*
*not contaminate the estimated parameters (fucoxanthin and peridinin ratios).*
and for that raison we flagged the pixels in the SeaWiFS images that have a *chl-a* concentration greater
than 3. mg m⁻³.

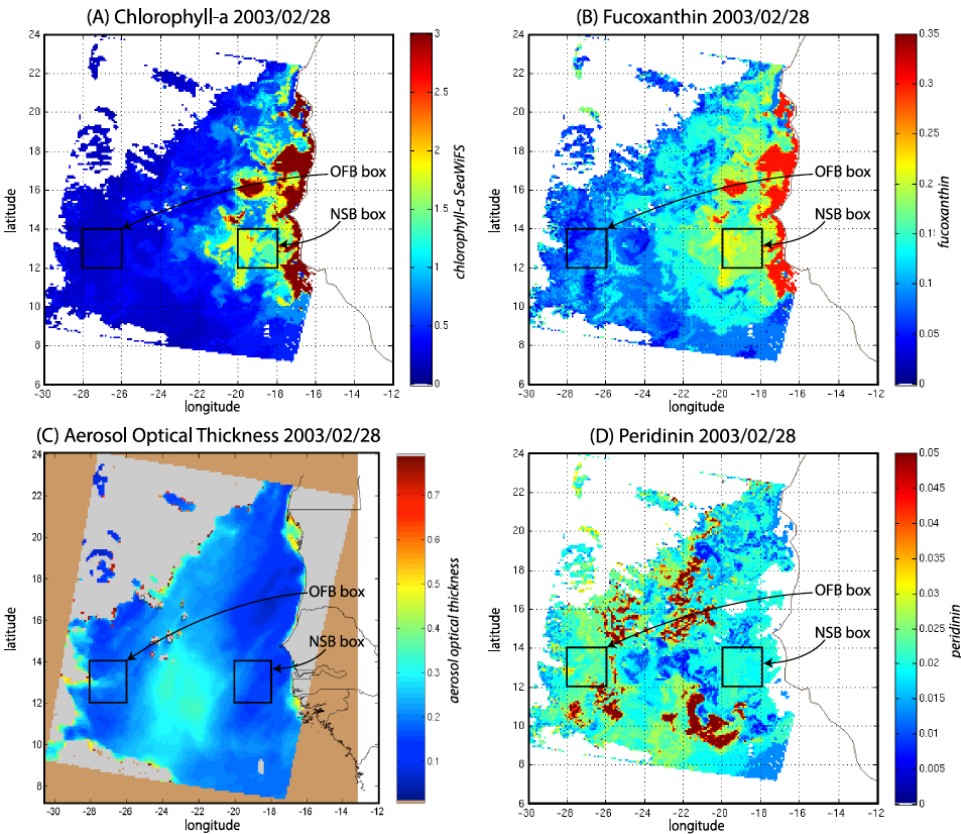



Figure 11: *(A) chl-a concentration, (B) fucoxanthin ratio, (C) aerosol optical thickness, (D) Peridinin for*
*28 February 2003. Panels (B) and (D) show that a second order information was retrieved, which is*
*correlated with the chl-a concentration (A) but is not equivalent. It is found that the aerosol optical*
*thickness (C) does not contaminate the estimated parameters (fucoxanthin and peridinin ratios). The*
*position of the NSB and OFB boxes are figured out by black square boxes.*

Regarding the images obtained for 1 January 2003 in the Senegalo-Mauritanian region
(Fig 8A, B, C, D), we observe that the *chl-a* (Fig 8A) is very high at the coast and decreases offshore
in accordance with the upwelling intensity as shown in the SST image (Fig 9). Moreover, we observed
a persistent well-marked *chl-a* pattern south of the Cap Vert peninsula in form of a "W", which is the
signature of a baroclinic Rossby wave (*Sirven et al*, 2019).
Except in the southern part of the region, the AOT (Aerosol Optical Thickness) is low, which means
that the atmospheric correction of the reflectance is quite small, which gives confidence in the ocean-
color data products. The fucoxanthin concentration is maximum at the coast and decreases offshore as
does the *chl-a* concentration, in agreement with the works of *Uitz et al.,* (2006, 2010). Fucoxanthin
presents coherent spatial patterns. Peridinin concentration is somewhat complementary to that of
fucoxanthin, with the low fucoxanthin concentration area corresponding to high peridinin
concentration area (northern part of Figs 8B, D). This behavior is also observed in Figure 10 (6 January
2003) and in Figure 11 (28 February, 2003) endorsing the analysis shown in Figure 8.
For 28 February, we selected two square box regions (Fig. 11), one near the coast (NSB,
long [-20°, -18°], lat [12°,14°]) and the other about 800 km offshore (OFB, long [-28°, -26°], lat
[12°,14°]). NSB waters correspond to upwelling waters while OFB waters correspond to oligotrophic
waters. We projected the eleven ocean color parameters of the NSB and OFB pixels on the 2S-SOM
map.

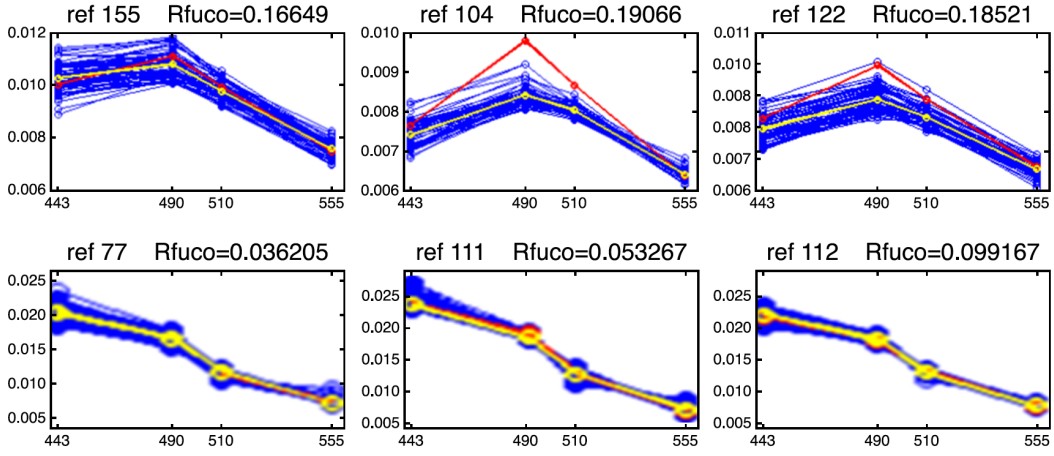

Figure 12: *Reflectance spectra (in blue) captured the 28 February by six neurons whose referent vector*
*spectra are in yellow: top line, for pixels in the NSB region (long. [-20°, -18°], lat. [12°, 14°]); bottom*
*line, for pixels in the OFB region (long. [-28°, -26°], lat. [12°, 14°]).*


Figure 12 presents the reflectance spectra (in blue) captured by three neurons of the 2S-SOM
corresponding to pixels located in the NSB region (*top line*) and those captured by three neurons
corresponding to pixels located in the OFB region (*bottom line*). The reflectance spectra of the
associated referent vectors *w* are in yellow. The satellite reflectance spectra match the referent vector
spectra; moreover the fucoxanthin ratio varies inversely with the mean value of the spectrum: the
higher the fucoxanthin ratio, the smaller the mean value of the spectrum. The pigment concentration
is greater near the coast.
We note a strong difference between the shape and the intensity of the near-shore (NSB) and offshore
(OFB) spectra. The OFB spectra present mean values higher than those of the NSB spectra. This is
due to the fact that NSB spectra were observed in a region where diatoms are abundant, as shown by

the high value of fucoxanthin concentration in this region (Figs 8, 10, and 11), which is a proxy for diatoms along with higher *chl-a* concentration. In Figure 12, we note the lower values of the coastal spectra at 443 nm, which can be interpreted as a predominant effect of spectral absorption by phytoplankton pigments and CDOM. The different spectra are close together in the OFB region and more disperse in the NSB region. This can be explained by the fact that the OFB region corresponds to Case-1 waters while the NSB region waters are close to Case-2 waters and are influenced by the variability of near shore process like turbidity or presence of dissolved matters, and dynamical instabilities.

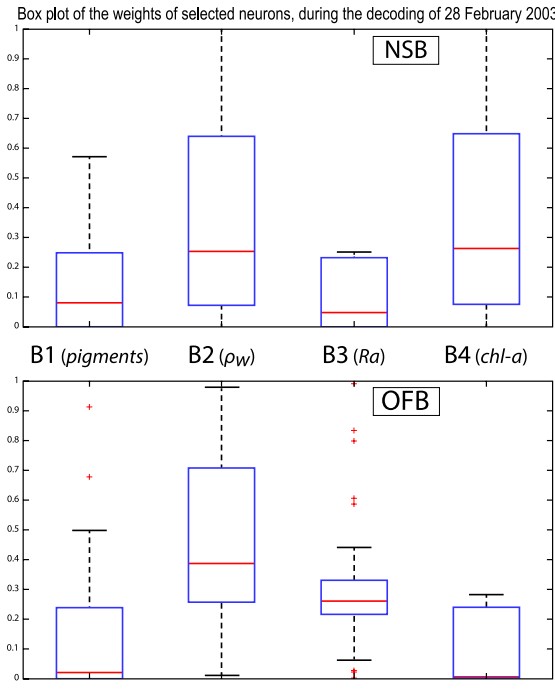

Figure 13: *Box plot of the weights of the selected neurons during the decoding of the 28 February data. From left to right, weights of blocks B1, B2, B3, B4. Top panel, in the NSB region (long. [-20°, -18°], lat. [12°, 14°]); bottom panel, in the OFB region (long. [-28°, -26°], lat. [12°, 14°]).*

We analyzed the weights of the blocks for the neurons selected in the analysis of the costal (NSB) and offshore (OFB) boxes. Figure 13 presents the box plot of the weight $\alpha_{cb}$ corresponding to the neurons belonging to the four blocks (B1, B2, B3, B4), with the constrain that the sum of the weights of a neuron is 1; a weight $\alpha$ larger than 0.25 indicates the predominance of a block in the learning for the classification (see section 3.5). It is clear that the weights for pixels near the coast (Fig 13, top panel) are different from those for offshore pixels (Fig. 13, bottom panel). As already mentioned in section 4.3 and also shown in Figure 7, the weights of the 2S-SOM play a significant role in the 2S-SOM

topology and consequently in the pigment retrieval. The weights of blocks B1 and B4 that take into account the influence of the pigment ratios and the chlorophyll content in the retrieval are very low for the offshore (OFB) oligotrophic region and more important for the coastal (NSB) region. The weights of the blocks B2 and B3, which take into account the influence of the reflectance ($\rho_w(\lambda)$, $Ra(\lambda)$), dominate for the offshore regions. In coastal waters, the weights of all the blocks are used, with a smaller influence of B3, which is associated with $R_a$. This gives information on the role played by the different variables on the classification in waters having different phytoplankton concentration and composition. Besides it shows the automatic adaptation of the 2S-SOM to the environment in order to optimize the clustering efficiency with respect to a classical SOM.

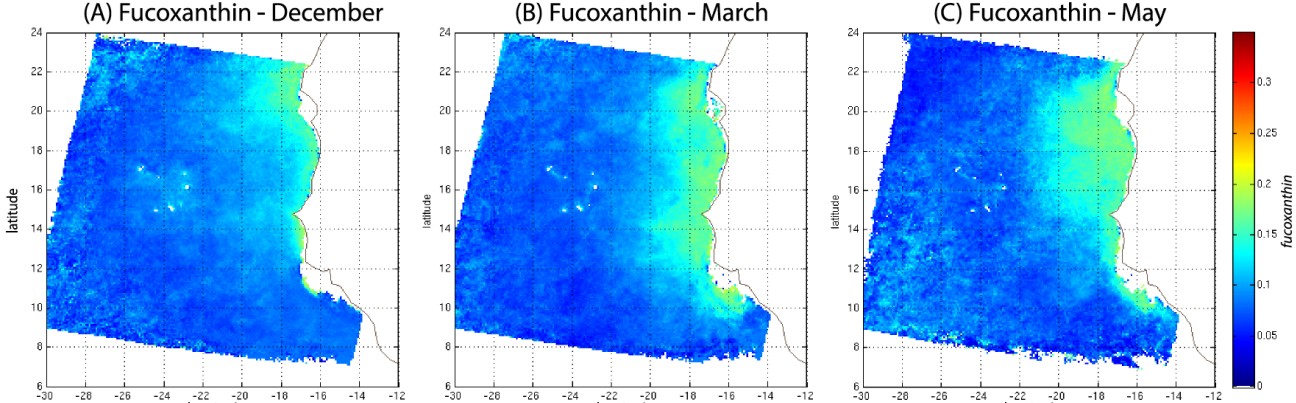

Figure 14: *Monthly fucoxanthin concentration averaged for an 11- years (1998-2009) for December (A), March (B) and May (C).*

In order to study the seasonal variability of the fucoxanthin concentration with some statistical confidence in the Senegalo-Mauritanian upwelling region, we constructed a monthly climatology for an 11-year period (1998–2009) of the SeaWiFS observations by summing the daily pixels of the month under study. The resulting climatology is presented in Figure 14 for December (Fig. 14a), March (Fig. 14b), and May (Fig 14c), which correspond to the most productive period (Fig. 14c). The fucoxanthin concentration, and consequently the associated diatoms, presents a well-marked seasonality. Fucoxanthin starts to develop in December North of 19°N, presents its maximum intensity in March when the upwelling intensity is maximum, extends up to the coast of Guinea (12°N) in April and begins to decrease in May where it is observed north of Cabo Verde peninsula (15°N) in agreement with the observations reported by *Farikou et al,* (2015) and *Demarcq and Faure,* (2000).

Figure 15 shows the fucoxanthin (in green) and the *chl-a* (in blue) concentrations computed from satellite observations for an 11-year period of SeaWiFS observations in the NSB region. There is a good correlation in phase between these two variables but not in amplitude (a good coincidence of

peak occurrence but weak correlation in peak amplitude) showing that the relationship between

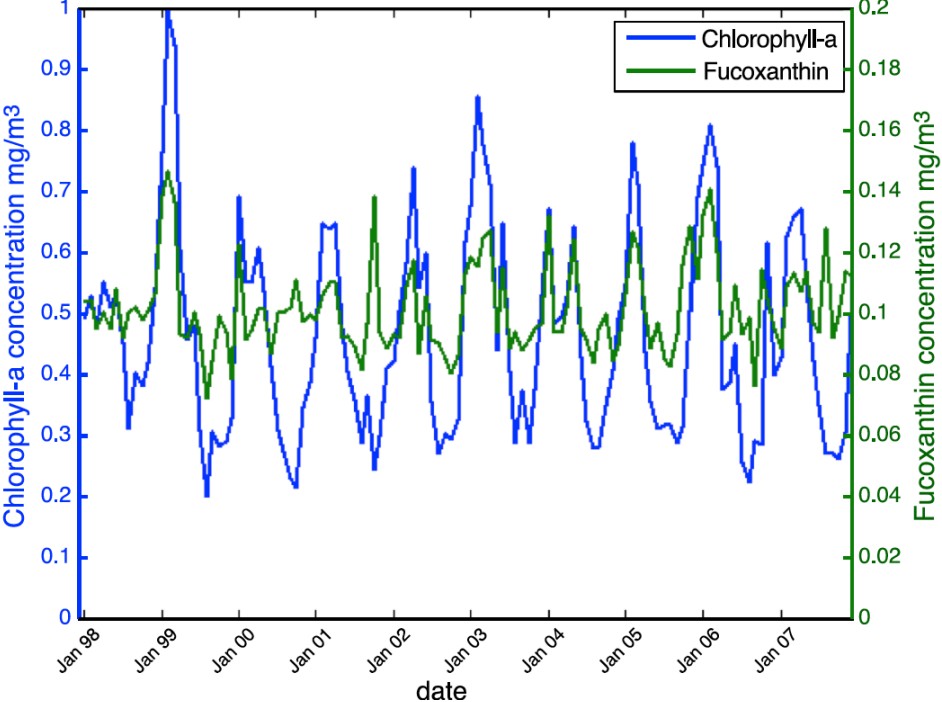

Figure 15: *. chl-a (in blue) and fucoxanthin (in green) concentrations for near-shore pixels (in the NSB region).*

fucoxanthin and *chl-a* is complex as mentioned by *Uitz et al*, (2006). In particular, there is a weak peak in fucoxanthin in October 2001, which is not correlated with a *chl-a* peak.

### 5-2 Analysis of the UPSEN campaigns

Figure 16 shows, for every UPSEN stations 1, 2, 3, 5a and 5b (see figure 1 for their geographical position), the averaged in-situ UPSEN spectrum (in blue), the referent spectrum (in red) of the 2S-SOM neuron captured by the collocated satellite VIIRS sensor observations. The referent spectrum is the mean of the different spectra captured by that neuron during the learning phase. Among these different spectra, there is one (black curve in figure 16) which is the closest to the UPSEN spectrum. Obviously, the black curve is closer to the blue curve than the red one which is flatten due to the averaging process. These three spectra are close together showing the good functioning of the 2S-SOM.

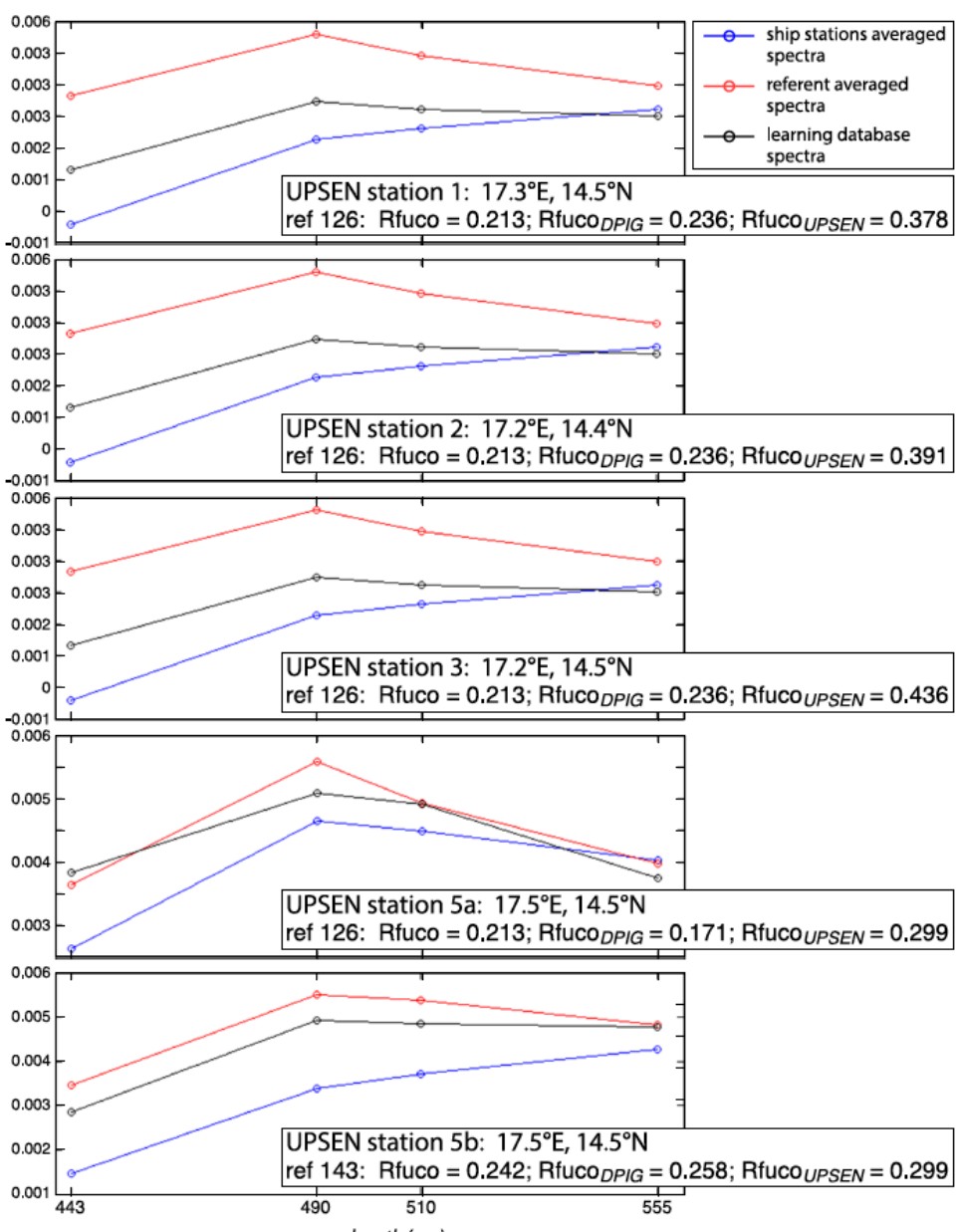

Figure 16: *For ship stations 1, 2, 3, 5a and 5b, we show the averaged spectrum of the in situ spectra of the UPSEN station in blue; the spectrum of the referent vector (in red) of the 2S-SOM neuron, which has captured the closest satellite observations to the UPSEN station; among the different spectra constituting the referent spectrum, the spectrum of the learning database (DGIP) that is the closest to the averaged satellite spectra is shown in black. In the rectangular cartoons, we show the position of the UPSEN station, the number of the neuron of the 2S-SOM which has captured the satellite observation, the Rfuco of the referent vector, the Rfuco$_{DGIP}$ of the closest DGIP and the in situ Rfuco$_{UPSEN}$.*

Their shapes are close to these observed in the NSB region (Figure 12) but their intensity is lower meaning that their waters are more absorbing than the NSB waters due to a higher pigment concentration. In fact, the UPSEN stations were located close to the coast (figure 1) in the Hann bight south off the Cap Verde peninsula, which is very rich in phytoplankton pigments. In table 3, we present the fucoxanthin ratios associated with the referent vectors ($Rfuco_{2S-SOM}$), the closest DPIG fucoxanthin-ratios captured by the neuron of the referents and the fucoxanthin-ratios measured during the UPSEN campaign. We note that the fucoxanthin ratios of the in-situ measurements are in the range of the DPIG (see table 1), which allows a good functioning of the 2S-SOM estimator. The pigment ratios obtained from ocean-color observations through the 2S-SOM are close to pigment concentrations measured at the ship stations, which confirms the validity of the method we have developed. We remark that the best 2S-SOM estimate of fucoxanthin ratio with respect to the UPSEN in-situ measurement is given at station 5b which is the farthest off the coast. These results endorse the climatological study of the Senegalo-Mauritanian upwelling region we have done with the 2S-SOM (section 5.1).

| UPSEN STATION | REFERENT N° | RFUCO 2S-SOM | RFUCO DPIG | RFUCO UPSEN |
|---|---|---|---|---|
| STAT 1   17.3E  14.5 N | 126 | 0.213 | 0.236 | 0.378 |
| STAT 2   17.2E  14.4 N | 126 | 0.213 | 0.236 | 0.391 |
| STAT 2   17.2E  14.5 N | 126 | 0.213 | 0.236 | 0.436 |
| STAT 5A  17.5E  14.5 N | 126 | 0.213 | 0.171 | 0.299 |
| STAT 5B  17.5E  14.5 N | 143 | 0.242 | 0.258 | 0.295 |

Table 3: *For ship stations 1, 2, 3, 5a and 5b of the UPSEN campaigns, we show the referent captured by the VIIRS observations, the fucoxanthin-ratio associated with this referent (Rfuco-2S-SOM), the fucoxanthin-ratio of the closest DPIG fucoxanthin-ratio captured by the neuron of the referent and the fucoxanthin-ratio measured in situ during the UPSEN campaign*

The 2S-SOM method gives pigment concentrations that are close to those obtained by in situ observations. The method could be applied to a large variety of other parameters in the context of studying and managing the planet Earth. The major constraint to obtaining accurate results is to deal with a learning data set that statistically reflects all the situations encountered in the observations processed. Due to its construction, the method cannot be used to find values beyond the range of the learning data set.

**6 - DISCUSSION**

Machine learning methods are powerful methods to invert satellite signals as soon as we have adequate database to support the calibration. Several technics have been used for retrieving biological information from ocean color satellite observations. First, studies employed multilayer perceptrons (MLP), which are a class of neural networks suitable to model transfer function (*Thiria* et al, 1993). *Gross* et al, (2000, 2004) retrieved *chl-a* concentration from SeaWiFS, *Bricaud* et al, (2006) modeled the absorption spectrum with MLP, *Raitsos* et al, 2008 and *Palacz* et al, 2013 introduced additional environmental variables in their MLPs such as SST in the retrieval of PSC/PFT from SeaWiFS, which improved the skill of the inversion. Another suitable procedure was to embed NN in a variational inversion, which is a very efficient way when a direct model exists (*Jamet* et al, 2005; *Brajard* et al, 2006a,b; *Badran* et al, 2008). Statistical analysis of absorption spectra of phytoplankton and of pigment concentrations were conducted by *Chazottes* et al, (2006, 2007), by using a SOM.

In the present study, due to the fact that the learning dataset was quite small (515 elements), we used an unsupervised neural network classification method, which is an extension of the SOM method well adapted to dealing with a small database whose elements are very inhomogeneous. We clustered available satellite ocean-color reflectance at five wavelengths and their derived products, such as chlorophyll concentration, and the associated in situ pigment ratios.

The major points of this study are as follows:

- The clustering was carried out by developing a new neural classifier, the so-called 2S-SOM, which presents several advantages with respect to the classical SOM. As in the SOM, we defined clusters that assemble vectors, which are close together in terms of a specified distance. This classifier was learned from a worldwide database (DPIG) whose vectors are ocean-color parameters observed by satellite multi-spectral sensors and associated pigment concentrations measured in situ. In the operational phase, SeaWiFS images are decoded, allowing the estimation of the pigment concentration ratios. The major advantage of 2S-SOM with respect to the classical SOM is to cluster variables having similar physical significance in blocks having specific weights. The weights attributed to the four blocks are computed during the learning phase and vary with the quality of the variables and with respect to their location on the ocean (near the coast or offshore). This permits to modulate the variable influence in the cost function, which makes the clustering more informative than that provided by the SOM. The block decomposition provides useful scientific information. For offshore, the weight analysis allowed us to show that more influence is given to the reflectance ratios $Ra(\lambda)$ and less to the *chl-a* and pigment concentrations; on the contrary near the coast the weights

indicate a more active use of the pigment composition and the *chl-a* concentration. Therefore, the
resulting 2S-SOM clustering therefore at best takes into account the information that belongs to the
specific water content.
- The 2S-SOM decomposes the DPIG into a large number of significant ocean-color classes allowing
reproduction of the different possible situations encountered in the dataset we analyze. Besides, we
assume that the relationship between the pigment concentration and the remote sensed ocean-color
observations is independent on the location, which is justifiable since the relationship depends on the
optical properties of ocean waters through well-defined physical laws which are region-independent.
This also endorses the fact that we used a global database to retrieve pigments in a definite region.
On the contrary, the different phytoplankton species vary from one region to another making the
relationship between pigment ratio and phytoplankton species strongly depending on the region. This
justifies the fact we focused our study on the pigment retrieval rather than on the PSC or PFT, as
mentioned above. Moreover, most of the recent phytoplankton in situ identifications have been made
using pigment measurements with the HPLC method (*Hirata et al*, 2011). It is therefore more natural
to retrieve the pigment concentrations, which is the quantity we measured, than the associated PSC
or PFT, which are estimated from the pigment observations through complex non-linear and region-
dependent algorithms (*Uitz et al*, 2006). Due to the characteristics of the DPIG, the method can
retrieve pigment concentration patterns over a large range ($0.02 - 2$ mg m$^{-3}$).
- We were able to analyze the pigment concentration in the Senegalo-Mauritanian region by processing
satellite ocean color observations with the 2S-SOM. We found an important seasonal signal of
fucoxanthin concentration with a maximum occurring in March. We evidenced a large offshore
gradient of fucoxanthin concentrations, the near shore waters being richer than the offshore ones. We
showed that the offshore region waters correspond to Case-1 waters, while the near shore waters are
close to Case-2 waters and are influenced by the variability of near shore process like turbidity, or
the presence of dissolved matters. The UPSEN measurements show that the pigment ratios of the
Senegalo-Mauritanian region are in the range of the DPIG database used to calibrate the method,
which justifies the use of the 2S-SOM algorithm to investigate this region.
- We used daily satellite observations to construct a monthly climatology of pigment concentrations
of the Senegalo-Mauritanian upwelling region, which has been poorly surveyed by oceanic cruises.
Due to the highly non-linear character of the algorithms for determining the pigment concentrations
from satellite measurements, it is mathematically more rigorous to apply these algorithms to daily
satellite data and to average this daily estimate for the climatology period under study, than to
estimate them from the satellite data climatology, as many authors have done (*Uitz et al., 2010*;
*Hirata et al.,* 2011). We found that Fucoxanthin starts developing in December North of 19°N,
presents its maximum intensity in March when the upwelling intensity is maximum, extends up to
the coast of Guinea (12°N) in April and begins to decrease in May

Another important aspect of our study concerns the validity of our results. The 2S-SOM method has
been validated by focusing the retrieval accuracy on the fucoxanthin ratio, by using a cross-validation
procedure. These results were qualitatively confirmed by two other independent studies.
- We first applied a cross validation procedure (see section 4.1), which is powerful technique for
validating models (*Kohavi,* 1995; *Varma* and *Simon*, 2006). We learned 30 different 2S-SOM using
30 different learning dataset determined at random from the DPIG dataset (each learning dataset
representing 90% of DPIG) and 30 test datasets (10% of DPIG). By averaging the results, we found
that the 2S-SOM method retrieves the fucoxanthin concentration with a good score (see the
statistical parameters in table 2) which confirms the pertinence of the method.
- We then found that our fucoxanthin climatology is in agreement with in situ observations of
phytoplankton reported in *Blasco et al*. (1980) in March to May 1974 off the coast of Senegal during
the JOINT I experiment. These authors analyzed 740 water samples collected with Niskin bottles
at 136 stations extending along a line at 21°40'N (in the northern part of the studied region) from 0
to 100 km offshore. The samples were taken at several depths (mostly at 100, 50, 30, 15, 5 m).
Phytoplankton cells were counted and identified by the Utermohl inverted microscope technique
(*Blasco,* 1977). These authors found that diatoms reach their maximum concentration in April–May
and are the most abundant group in that period, whereas the other cells predominate in March.
Similar microscope observations have been reported in the ocean area south of Dakar by *A. Dia*
(1985) during several ship surveys in February–March 1982–1983.
- Our method is also in agreement with the monthly eleven years climatology presented in *Farikou et*
*al,* (2015) who used a modified PHYSAT method to retrieve the *PFT* in the Senegalo-Mauritanian
region.
- The pigment concentrations provided by the 2S-SOM from the VIIRS sensor observations are in
qualitative agreement with the in-situ measurements done at five stations during the two UPSEN
campaigns in 2012 and 2013, showing that the method is able to function in waters where the
pigment concentrations are quite high (fucoxanthin ratios of the order 0.4).





**7 - CONCLUSION**

We developed a new neural network clustering method, the so-called 2S-SOM algorithm to retrieve
phytoplankton pigment concentration from satellite ocean color multi spectral sensors. The 2S-SOM
algorithm is a SOM specifically designed to deal with a large number of heterogeneous components
such as optical and chemical measurements. The major advantage of 2S-SOM with respect to the
classical SOM is to cluster variables having similar significance in blocks having specific weights.
The weights attributed to the blocks during the learning phase vary with the quality of the variables in
the classification. This permits to modulate the variable influence in the cost function, which makes
the clustering more informative than that provided by the SOM. Besides, the block weighting provides
useful information on the functioning of the classification by permitting to identify the variables which
control it. It also allows us to better understand the dynamics of the phytoplankton communities.
The 2S-SOM method is efficient and rapid as soon as the calibration is done, since it uses elementary
algebraic operations only. The 2S-SOM method is like a piecewise regression that takes advantage of
the unsupervised classification of the SOM. We decomposed the DPIG database into quite a large
number of partitions (9x8=162) when comparing our study to other studies (*Uitz et al*, 2006, 2012).
The validity of the method has been controlled through a cross validation procedure and confirmed by
three qualitative studies. Statistical parameters ($R^2$ coefficients, RMSE and P-values) of the cross-
validation between the DPIG in situ pigments and the pigments given by the 2S-SOM averaged for the
30 2S-SOM realizations presented in table 2, show the good performance of the method. It must be
noticed that the performance mainly depends on the size of the learning set used to calibrate the 2S-
SOM. This set must include all the situations encountered in the pigment retrieval. The larger the
learning set, the better the method performs. Due to its generic character and its flexibility, the method
could be used to determine a large variety of measures done with satellite remote sensing
observations.
In this work, the method was applied to study the seasonal variability of the fucoxanthin concentration
in Senegalo-Mauritanian upwelling region. We showed a large offshore gradient of fucoxanthin, the
higher concentration being situated near the shore. We were able to construct a monthly climatology
for an 11-year period (1998–2009) of the SeaWiFS observations by summing the daily pixels of the
month under study in a region which was poorly surveyed by oceanic cruises. The fucoxanthin
concentration, and consequently the associated diatoms, present a well-marked seasonality (Figure 10).
Fucoxanthin starts developing in December North of 19°N, presents its maximum intensity in March
when the upwelling intensity is maximum, extends up to the coast of Guinea (12°N) in April and
begins to decrease in May where it is observed north of Cabo Verde peninsula (15°N), in agreement

732 with the observations reported by *Farikou et al,* (2015) and *Demarcq and Faure*, (2000). The UPSEN

733 campaign results endorse the validity of the study of the Senegalo-Mauritanian upwelling region done

734 with the 2S-SOM.


736 **Acknowledgments**

737 The study was supported by the projects CNES-TOSCA 2013-2014 and 2014-2015. The water-leaving

738 reflectances were obtained from the SeaWiFS daily reflectances, $\rho obsTOAw(\lambda)$, provided by the

739 NASA/GSFC/DAAC observed at the top of the atmosphere (TOA) and processed with the SOM-NV

740 algorithm (Diouf et al., 2013) from 1998 to 2010. They are available at the web site:

741 http://poacc.locean-ipsl.upmc.fr/. The DPIG data base was kindly provided by Dr. S. Alvain. We thank

742 Dr. Alban Lazar and Dr. E. Machu for providing in situ data measured during the UPSEN experiments

743 as well as stimulating discussions for their interpretation. We also thank Ray Griffiths for editing the

744 manuscript.

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

Jeffreys S.W. and Vesk M. : Phytoplankton Pigment in Oceanography : Guidelines to Modern
Methods, UNESCO,  Paris, ed S. W. Jeffery,  R.F.C. Mantoura and S. W. Wright, Introduction to
marine phytoplankton and their pigment signatures, pp 33-84, 1997.
Jouini M., Lévy M. , Crépon M.  and Thiria S. : Reconstruction of ocean color images under clouds
using a neuronal classification method. Remote Sens. Environ. vol **131**, pp 232-246, 2013
Kohavi R. : A study of cross-validation and bootstrap for accuracy estimation and model selection.
Proceedings of the Fourteenth International Joint Conference on Artificial Intelligence. San Mateo,
CA: Morgan Kaufmann ed.. **2** (12): pp 1137–1143, 1995.
Kohonen T : Self-organizing maps (3$^{rd}$ ed.). Springer, Berlin Heidelberg New York. 2001
Kruizinga S. and Murphy A : Use of an analogue procedure to formulate objective probabilistic
temperature forecasts in the Netherlands. Mon. Wea. Rev., vol **111,** pp 2244–2254, 1983.
Le Quéré et al, (2018) : Global Carbon Budget 2018, Earth Syst. Sci. Data, 10, 2141–2194, 2018 ;
https://doi.org/10.5194/essd-10-2141-2018
Lévy M., D. Iovino, L. Resplandy, P. Klein, G. Madec, A.-M. Tréguier, S. Masson, K. Takahashi, Large-scale
impacts of submesoscale dynamics on phytoplankton: Local and remote effects, Ocean Modelling, 77–93,

2012

Levy, M., Mesoscale variability of phytoplankton and of new production: Impact of the large-scale nutrient
distribution, J. Geophys. Res., 108(C11), 3358, doi:10.1029/2002JC001577, 2003.
Liu Y. and Weisberg R. H. : Patterns of ocean current variability on the West Florida Shelf using the
self-organizing map, J. Geophys. Res., **110,** C06003, doi:10.1029/2004JC002786, 2005
Liu Y., Weisberg R. H., and He R. : Sea surface temperature patterns on the West Florida Shelf using
growing hierarchical self-organizing maps, J. Atmos. Oceanic Technol., vol **23**(2), pp 325– 338, 2006
Longhurst A. R., Sathyendranath S., Platt T., Caverhill C. : An estimation of global primary production
in the ocean from satellite radiometer data. J. Plank. Res. vol **17**, pp 1245-1271, 1995
Lorenz E. N : Atmospheric predictability as revealed by naturally occurring analogs. J. Atmos. Sci.,
vol 26, pp 639–646, 1969
Morel A. and Gentili G. : Diffuse reflectance of oceanic waters. III. Implication of bidirectionality for
the remote-sensing problem. Appl. Opt. vol 35, pp 4850-4862, 1996.
Mouw C. B. and Yoder J. A. : Optical determination of phytoplankton size composition from global
SeaWiFS imagery. J. Geophys. Res. vol **115**, C12018, doi:10.1029/2010JC006337, 2010.
Ndoye S. , Capet X., Estrade P., Sow B., Dagorne D., Lazar A., Gaye A. and Brehmer P. : SST patterns
and dynamics of the southern Senegal-Gambia upwelling center. J. Geophys. Res. Oceans, vol 119,
pp 8315–8335. 2014
Niang, A., Gross, L., Thiria, S., Badran, F., & Moulin, C. Automatic neural classification of ocean
colour reflectance spectra at the top of atmosphere with introduction of expert knowledge.
Remote Sens. Environ, vol 86, pp 257–271, 2003.
Niang A., Badran F., Moulin C., Crépon M. and Thiria S. : Retrieval of aerosol type and optical
thickness over the Mediterranean from SeaWiFS images using an automatic neural classification
method. Remote Sens. Environ. vol 100, pp 82-94, 2006.
O'Reilly, J.E., Maritorena , S., Siegel, D. A., O'Brien, M. C ., Toole, D., Mitchell, B. G., Kahru, M.,
Chavez, F. P., Strutton, P., Cota, G. F., Hooker, S. B., McClain, C. R., Carder, K. L., Muller-
Karger, F., Harding, L., Magnuson , A., Phinney, D., Moore, G.F., Aiken, J., Arrigo, K. R.,
Letelier, R., and Culver, M.    Ocean color chlorophyll a  algorithms for SeaWiFS, OC2 and
OC4: Version 4. In S. B. Hooker, and E. R. Firestone (Eds), *SeaWiFS postlaunch calibration and*
*validation analyses: Part 3. NASA Tech. Memo. 2000-206892, vol. 11*(pp.9-23). Greenbelt, MD:
NASA Goddard Space Flight  Center. 2001.
Palacz A. P., St. John, M. A., Brewin, R. J.W., Hirata, T., and Gregg,W.W. : Distribution of
phytoplankton functional types in high-nitrate low-chlorophyll waters in a new diagnostic
ecological indicator model. Biogeosciences 10, 7553–7574. doi: 10.5194/bg-10-7553, 2013.
Raitsos D. E., Lavender, S. J., Maravelias, C. D., Haralambous, J., Richardson, A. J., and Reid, P.
C. : Identifying phytoplankton functional groups from space: an ecological approach. Limnol.
Oceanogr. 53, 605–613. doi: 10.4319/lo.2008.53.2.0605, 2008
Reusch D. B., Alley, R. B., and Hewitson, B. C : North Atlantic climate variability from a self-
organizing map perspective, J. Geophys. Res., vol **112**, D02104, doi:10.1029/2006JD007460, 2007.
Sathyendranath S., Watts S., L., Devred E., Platt T., Caverhill C. M., and  Maass H. :  Discrimination
of diatom from other phytoplankton using ocean-colour data, Mar. Ecol. Prog. Ser., vol 272, pp 59–

68, 2004.

Sirven J., Mignot J., Crépon M. : Generation of Rossby waves off the Cap Verde Peninsula: the role
of the coastline . Ocean Sci., 15, 1–24, 2019
Sosik, H.M.; Sathyendranath, S.; Uitz, J.; Bouman, H.; Nair, A. In situ methods of measuring
phytoplankton functional types. In Phytoplankton Functional Types from Space. IOCCG report, No.
15; Sathyendranath, S., Ed.; IOCCG: Dartmouth, NS, Canada, pp. 21–38, 2014.
Uitz J., Claustre H., Morel A. and. Hooker S.B : Vertical distribution of phytoplankton communities
in open ocean: an assessment based on surface chlorophyll. J. Geophys. Res. **111,** C08005,
doi:10:1029/2005JC003207. 2006
Uitz J., Claustre H., Gentili B. and Stramski D. : Phytoplankton class-specific primary production in
the world's ocean: seasonal and interannual variability from satellite observations. Global
Biogeochem. Cycles, vol **24**, GB 3016, doi:10:1029/2009GB003680, 2010
Van den Dool H. : Searching for analogs, how long must we wait? Tellus, vol **46A**, pp 314–324, 1994.
Varma, S., Simon, R. : Bias in error estimation when using cross-validation for model selection; BMC
Bioinformatics. vol **7**. PMC 1397873 . PMID 16504092. doi:10.1186/1471-2105-7-91, 2006
Vidussi F., Claustre H., Manca B. B., Luchetta A. and Marty J. C. : Phytoplankton pigment distribution
in relation to upper thermocline circulation in the eastern Mediterranean sea during winter. J.
Geophys. Res., vol 106, pp 19,939-19,956, 2001.
Westberry T., Behrenfeld M.J., Siegel D. A. and Boss E.: Carbon-based productivity modeling with
vertically resolved photoacclimatation. Global Biogeochem. Cycles, vol **22**, *GB2024*,
DOI:10.1029/2007GB003078, 2008
Zorita E. and Von Storch H. : The Analog Method as a Simple Statistical Downscaling Technique:
Comparison with More Complicated Methods. Journal of Climate, vol **12,** pp 2474-2489, 1999.

**ANNEX 1**

924   **A1  Cost function of the SOM**

925  Let us recall the following notation:

926  $D = \{z_1, \cdots, z_i, \cdots, z_K\}$ the dataset composed of $K$ vectors $z_i \in \mathbb{R}^N$

927  $W = \{w_1, \cdots, w_c, \cdots, w_C\}$ the set of weights $w_c \in \mathbb{R}^N$ where $C = p \times q$ is the size of the SOM.

928  The $w_c$ of the SOM are estimated by minimizing a cost function of the form

930   $$J_{SOM}^T(\chi, W) = \sum_{i=1}^{K} \sum_{c=1}^{p \times q} K^T\left(\delta\left(c, \chi(z_i)\right)\right) \|z_i - w_c\|^2, \qquad\qquad \text{(A.1)}$$

931  where $c$ indices the neurons of the SOM map, $\chi$ is the allocation function that assigns each element $z_i$

932  of $D$ to its referent vector $w_c$ which is of the form $\chi(z_i) = \arg\min_c \|z_i - w_c\|^2$,

933  $\delta\left(c, \chi(z_i)\right)$ is the discrete distance on the SOM between a neuron if index $c$ and the neuron allocated

934  to observation $z_i$ , and $K^T$ a kernel function parameterized by $T$ that weights the discrete distance on

935  the map and decreases during the minimization process. $T$ acts as a regularization term (*Kohonen,* 2001,

936  *Niang et al,* 2003). In the present case $K^T$ is of the form :

937  $K^T(\delta) = (1/T)K(\delta/T)$, where $K$ is the gaussian function of mean 0 and standard deviation 1.

938  The cost function (A.1) takes into account the proper inertia of the partition of the data set $D$ and

939  ensures that its topology is preserved.

941   **A2  Definition of the Algorithm 2S-SOM**

942  The 2S-SOM algorithm is an extension of the Self-Organizing maps (SOM, *Kohonen,* 2001) based on

943  the K-mean method (*Ouattara et al.*, 2014**,** https://www.theses.fr/179489704). It automatically

944  structures the variables having some common characters into conceptually meaningful and

945  homogeneous blocks during the learning phase. The 2S-SOM takes advantage of this structuration of

946  $D$ and the variables into $B$ different blocks, which permits an automatic weighting of the influence of

947  each block and consequently of each variable in the classification phase. The 2S-SOM is based on a

948  modification of the cost function of the SOM algorithm. For a neuron of index $c$, we define the weights

949  $\alpha_{cb}$ of each block $b$ ($b = 1, ..., B$) and the weights $\beta_{cbj}$ of the variables $j$ ($j = 1, ..., P_b$) in this block,

950  where $P_b$ is the number of variable in the block indexed by $b$. The vectors of weighs are denoted

951  $\boldsymbol{\alpha} = \{\alpha_{cb}\}_{1 \leq c \leq C, 1 \leq b \leq B}$ and $\boldsymbol{\beta} = \{\beta_{cbj}\}_{1 \leq c \leq C, 1 \leq b \leq B, 1 \leq j \leq P_b}$

952  The new cost function is:

$\quad J_{2S-SOM}^{T}(\chi, \mathbf{W}, \boldsymbol{\alpha}, \boldsymbol{\beta}) = \sum_{c} \left( \sum_{b=1}^{B} \left( \sum_{zi \in D} \alpha_{cb} K^{T}\left(\delta\left(c, \chi(z_i)\right)\right) d_{\beta_{cb}}(i) + J_{cb} \right) + I_c \right),$ $\quad$ (A.2)
$\quad$ with
$\quad d_{\beta_{cb}}(i) = \sum_{j=1}^{P_b} \beta_{cbj} \left(z_{ib}^{j} - w_{ib}^{j}\right)^2,$ $\qquad\qquad\qquad\qquad\qquad\qquad\qquad\qquad$ (A.3)
$\quad$ where $c$ indices the neurons of the 2S-SOM map.
$\quad$ under the two constraints:
$\quad \displaystyle\sum_{b=1}^{B} \alpha_{cb} = 1; \alpha_{cb} \in [0,1] \; \forall c, 1 \leq c \leq C$ $\qquad\qquad\qquad\qquad$ (A.4)
$\quad$ and
$\quad \displaystyle\sum_{j=1}^{P_b} \beta_{cbj} = 1; \beta_{cbj} \in [0,1], \forall c, 1 \leq c \leq C; \forall b, 1 \leq b \leq B.$
$\quad I_c$ and $J_{cb}$ are used to regularize the weights $\boldsymbol{\alpha}$ and $\boldsymbol{\beta}$. They are defined as negative entropies weighted
$\quad$ by $\mu$ for the blocks and $\eta$ for the variables of each block

$\quad I_c = \mu \displaystyle\sum_{b=1}^{P_b} \alpha_{cb} log(\alpha_{cb})$ $\qquad\qquad\qquad\qquad\qquad$ (A.6)
$\quad$ and
$\quad J_{cb} = \eta \displaystyle\sum_{j=1}^{B} \beta_{cbj} log(\beta_{cbj})$ $\qquad\qquad\qquad\qquad\qquad$ (A.7)
$\quad$ The topological conservation properties of 2S-SOM are influenced by the weights $\alpha_{cb}$ and $\beta_{cbj}$ in the
$\quad$ classification through the hyper-parameters $\mu$, $\eta$ and the neighborhood parameter T.
$\quad$ The weights $\alpha_{cb}$ and $\beta_{cbj}$ respectively indicate the relative importance of blocks and variables in the
$\quad$ neurons. Thus, the greater the weight of a block $b$ or a variable $j$, the more the block or the variable
$\quad$ contributes to the definition of the class (or neuron) in the sense that it makes it possible to reduce the
$\quad$ variability of the observations in the cell and in its close neighborhood. For a high value of $\eta$ and a
$\quad$ fixed one for $\mu$, the $\beta_{cbj}$ in a block are equal to $1/P_b$. In this case, only the blocks are modified according
$\quad$ to their capacity to define the neurons. In this context, the 2S-SOM then makes possible to weight the
$\quad$ different blocks for each neuron
-    For high values of $\mu$, $I_c$ is large. The minimization of $J_{cb}$ forces all its coefficients to become

equal. For a fixed value of $\eta$, the $\alpha_{cb}$ associated with the blocks are all equal to $1/B$. In this case,

only the $\beta_{cbj}$ of the variables inside the blocks weight the neurons

-    When $\mu$ and $\eta$ tend to very large values, the blocks are equiprobable as well as the variables.

Thus, the 2S-SOM algorithm is comparable to the SOM.


**A3 How the 2S-SOM algorithm works:**
For fixed $\mu$ and $\eta$, the learning of the 2S-SOM algorithm is as follows:
-    Step 0: Initialization with iteration of the algorithm SOM, by setting $\boldsymbol{\alpha}$ and $\boldsymbol{\beta}$ to homogeneous

values.

The optimization of $J^T_{2S-SOM}$ is carried out through an iterative process composed of three steps (1, 2,
and 3) presented below.
-    Step 1: The $\boldsymbol{w}_c$ referents, the weights $\alpha$ and $\beta$ are known and fixed, the observations are assigned

to the neurons by respecting the assignment function:

$$c(zi) = \chi(z_i) = \arg\min_{c \in C} \left( \sum_{r \in C} K^T\big(\delta(r,c)\big) \left( \sum_{b=1}^{B} \alpha_{cb} d_{\beta_{cb}}(i) \right) \right) \quad (A.8)$$


-    Step 2: Updating the neuron centers (the $\boldsymbol{w}_c$ referents) according to the formula of the SOM

algorithm.


-    Step 3: the assignment function and the referents $\boldsymbol{w}_c$ being fixed, $\boldsymbol{\alpha}$ and $\boldsymbol{\beta}$ are determined

according to the equations (A.9, A.10, A.11, A.12), by minimizing the cost function

$J^T_{2S-SOM}$ with respect to $\alpha$ and $\beta$ under the constraints (A.4) and (A.5).

$$\alpha_{cb} = \frac{\exp\left(\frac{-\psi_{cb}}{\mu}\right)}{\sum_{b=1}^{B} \exp\left(\frac{-\psi_{cb}}{\mu}\right)} \quad (A.9)$$

with

$$\psi_{cb} = \sum_{zi \in D} K^T\big(\delta(\chi(z_i), c)\big) d_{\beta_{cb}}(i) \quad (A.10)$$

and

$$\beta_{cbj} = \frac{\exp\left(\frac{-\Phi_{cbj}}{\eta}\right)}{\sum_{b=1}^{p_b} \exp\left(\frac{-\Phi_{cbj}}{\eta}\right)}$$ (A. 11)
with
$$\Phi_{cbj} = \sum_{z_i \in D} \alpha_{cb} K^T(\chi(z_i), c)(z_{ib}^j - w_{cb}^j)^2$$ (A. 12)

This algorithm is repeated by sampling the hyper-parameters $\mu$ and $\eta$ until convergence.
Finally, at the convergence, the 2S-SOM provides on the one hand a topological map allowing to
visualize the data, and on the other hand a weight system for the neurons of the map allowing us to
interpret the role of the different variables and to choose those that are the most significant for the
classification and to neutralize those which are the least significant.

**FIGURE CAPTION**

Figure 1 : *Mauritania and Senegal coastal topography. The land is in brown and the ocean depth is represented in meters by the color scale on the right side of the figure. The UPSEN stations are shown at the bottom left cartoon of the figure.*

Figure 2 : *Geographic positions of the 515 in situ and satellite collocated measurements of the DPIG database.*

Figure 3: *Dispersion diagram of DPIG chl-a computed from the SeaWiFS observations using the OC4V4 algorithm versus in situ chl-a. The coefficient of vraisemblance $R^2$ and the RMSE (Root Mean Square Error) were computed in mg $m^{-3}$*

Figure 4: *Flowchart of the method: top panel - Learning phase; bottom panel – operational phase which consists in pigment retrieval and the determination of the $\alpha_{cb}$ block parameters.*

Figure 5 : *Flowchart of the cross-validation procedure for 30 partitions of the DPIG database.*

Figure 6 : *2S-SOM Map. From left to right and top to bottom, values of the referent vectors for $\rho_w(490)$, Ra(490), SeaWiFS chl-a, and fucoxanthin, peridinin, divinyl Ratios. The number in each neuron indicates the amount of DPIG data captured at the end of the learning phase, the values indicated by the color bars are centered-reduced and non-dimensional values.*

Figure 7: *2S-SOM map. Weights ($\alpha_{cb}$) of the four block parameters determined at the end of the learning phase; from left to right and top to bottom: $\rho_w$, Ra, Pigment, SeaWifs chl-a. The color bars show the % of the weight estimated by 2S-SOM, a value of 1 or 0 indicating that the data in the neuron are assembled with respect to that block only.*

Figure 8 : *A) chl-a concentration, (B) fucoxanthin ratio, (C) aerosol optical thickness, (D) peridinin for 1 January 2003. Panels (B) and (D) show that a second-order information was retrieved, which is correlated with the chl-a concentration (A) but not equivalent. The aerosol optical thickness (C) does not seem to contaminate the estimated parameters (fucoxanthin and peridinin ratios).*

Figure 9 : *SST for 2 January 2003. Note the well-marked upwelling (cold temperature) north of 13°N.*

Figure 10 : *(A) chl-a concentration, (B) fucoxanthin ratio, (C) aerosol optical thickness, (D) peridinin for 6 January 2003. Panels (B) and (D) show that a second-order information was retrieved, which is correlated with the chl-a concentration (A) but is not equivalent. It is found that the aerosol optical thickness (C) does not contaminate the estimated parameters (fucoxanthin and peridinin ratios).*


Figure 11 : *(A) chl-a concentration,   (B) fucoxanthin ratio,   (C) aerosol optical thickness,*
*(D) Peridinin for 28 February 2003.  Panels (B) and (D) show that a second order information was*
*retrieved, which is correlated with the chl-a concentration (A) but is not equivalent.  It is found that*
*the aerosol optical thickness (C) does not contaminate the estimated parameters (fucoxanthin and*
*peridinin ratios). The position of the NSB and OFB boxes are figured out by black square boxes.*

Figure 12 : *Reflectance spectra (in blue) captured the 28 February by six neurons whose referent*
*vector spectra are in yellow: top line, for pixels in the NSB region (long. [-20°, -18°], lat. [12°,*
*14°]); bottom line, for pixels in the OFB region (long. [-28°, -26°], lat. [12°, 14°]).*

Figure 13    *Box plot of the weights of the selected neurons during the decoding of the 28 February*
*data. From left to right, weights of blocks B1, B2, B3, B4. Top panel, in the NSB region (long. [-20°,*
*-18°], lat. [12°, 14°]); bottom panel, in the OFB region (long. [-28°, -26°], lat. [12°, 14°]).*

Figure 14 : *Monthly fucoxanthin concentration averaged for an 11- years (1998-2009) for December*
*(A), March (B) and May (C).*

Figure 15 : .  *chl-a (in blue) and fucoxanthin (in green) concentrations for near-shore pixels (in the*
*NSB region).*

Figure 16 : *For ship stations 1, 2, 3, 5a and 5b, we show  the averaged spectrum of the in situ*
*spectra of the UPSEN stations in blue; the spectrum of the referent vector (in red) of the 2S-SOM*
*neuron, which has captured the closest satellite observations to the UPSEN station; among the*
*different spectra constituting the referent spectrum, the spectrum of the learning database (DGIP)*
*that is the closest to the averaged satellite spectra is shown in black. In the rectangular cartoons, we*
*show the position of the UPSEN station, the number of the neuron of the 2S-SOM which has*
*captured the satellite observation, the Rfuco of the referent vector, the $Rfuco_{DGIP}$ of the closest DGIP*
*and the in situ $Rfuco_{UPSEN}$.*


**Table Caption**

Table 1 : *Pigments of the DPIG and their statistical characteristics: STD (Standard Deviation), MIN*
*(minimum value), MAX (maximum value).*

Table 2 : *Statistical parameters ($R^2$ coefficients, RMSE and P-values) of the cross validation between*
*the DPIG in situ pigments and the pigments given by the 2S-SOM averaged for the 30 2S-SOM*
*realizations*

Table 3 : *For ship stations 1, 2, 3, 5a and 5b of the UPSEN campaign, we show the referent captured*
*by the VIIRS observations, the fucoxanthin-ratio associated with this referent (Rfuco-2S-SOM), the*
*fucoxanthin-ratio of the closest DPIG fucoxanthin-ratio captured by the neuron of the referent and the*
*fucoxanthin-ratio measured in situ during the UPSEN campaign.*

**Author Contribution**
Dr N'Dye Niang and Maurice Ouattara provided the 2S-SOM code, Khalil Yala processed the data
and did the computations with the 2S-SOM, Sylvie Thiria, Michel Crepon and Julien Brajard
analyzed the results, Carlos Mejia and Roy El Hourany did the statistical tests presented in tables and
figure 13. Prof. Sylvie Thiria conceived and supervised the study.


**Code/data availability**
The satellite data (ocean color and SST) are available at the web site:
http://poacc.locean-ipsl.upmc.fr/.
The DPIG data base was kindly provided by Dr. S. Alvain (Severine.alvain@univ-littoral.fr)
The UPSEN data are available at : alban.lazar@locean-ipsl.upmc.fr
The 2S-SOM code is available on request at:  carlos.mejia@locean-ipsl.upmc.fr


**Short summary**
The paper is a contribution to the study of the phytoplankton pigment climatology from satellite
ocean colour observations in the Sénégalo-Mauritanian upwelling, which is a very productive region
where in situ observations are lacking. We processed the satellite data with an efficient new neural
network classifier. We were able to provide the climatological cycle of diatoms. This study may have
an economic impact on fisheries thanks to a better knowledge of phytoplankton dynamics.