# Peer review of "ESTIMATION OF PHYTOPLANKTON PIGMENTS FROM OCEAN-COLOR SATELLITE OBSERVATIONS IN THE SENEGALO-MAURITANIAN REGION BY USING AN ADVANCED NEURAL CLASSIFIER By Khalil Yala1, N'Dye Niang2, Julien Brajard1,4, Carlos Mejia1, Maurice Ouattara2, Roy El Hourany1, Michel Crépon1 and Sylvie Thiria1,3 1 IPSL/LOCEAN, Sorbonne Université (Université Paris6, CNRS, IRD, MNHN), 4 Place Jussieu, 75005 Paris, France 2 "

_Ocean Science, 2019_

## Referee Comment (RC1) · Yangyang Liu (Referee) · 2 May 2019

General comments:

Measurement of oceanic phytoplankton pigments plays a crucial role in understanding biological response to global climate change. This study proposed a Self Organized Map (2S-SOM) to estimate the pigment concentrations from satellite ocean colour products in the SeÌ Ąne Ì Ągalo-Mauritanian region. It is an interesting and important study to improve our understanding on the existing remote sensing methods.

[Figure]

Specific comments: 1. The introduction can be written in a much more accurate way. I would check it phrase by phrase and sentence by sentence. Take phytoplankton identification method as an example: 1) Line 50-52: For example, one limitation of microscopy is the difficulty in indentifying picoplankton. 2) The optical microscopy method is developing, for example the imaging flow cytometry (IFC). 3) Line 54-55: Mind the use of the terms PSC and PFT. PFT depends on how you define it. PSC is also a type of PFT definations. 4) Line 57-60: the conversion formula method is the so called "Diagonstic Pigment Analysis". CHEMTAX uses matrix factorization to estimate PFT from pigments. 5) Line 60: I am not sure with just marker pigments themselves the identification of phytoplankton can be achieved in species level. 6) In summary, please check IOCCG report 15 and related literature carefully. 2. Line 139-140. Match-up procedure can be more detailed, for example, by adding the ceiteria of refusing data points and the reason why you choose 20km. 3. Line 150-160 and Figure 3. Please use more statistical metrics in addition to R-square and RMSE according to Brewin et al 2015. Please specify whether they are calculated in log scale or not. Brewin, Robert JW, et al. "The Ocean Colour Climate Change Initiative: III. A round-robin comparison on in-water bio-optical algorithms." Remote Sensing of Environment 162 (2015): 271-294. 4. Line 288-289: you have said the same as Line 264-265. 4. Table 2: often these statistics are done on log(pigments) - given their distribution and expected errors. 5. Line 402: Unfortunately it cannot be concluded that diatoms dominated because of high Fuco ratio and chl-a, without additional information on phytoplankton groups using e.g. microscopy. 6. Please spell MLP out in the Discussion section. 7. Line 649-654: Can you summarize why SOM needs fewer data points than MLPs and other supervised learning? Why MLP cannot be trained with total $\sim$500 dapa points? 8. Is it possible to clarify the minimum threshold of pigment concentration of the applicability of 2S-SOM?

Technical corrections: 1. The country Senegal has three versions of names in the manuscript, i.e. SeÌAn̦eÌA̦galo (title), Senegalo (context) and Senegal (Figure 1). Please keep the consistency. 2. Line 41: The word "phytoplankton" is more often

used as plural. 3. Line 42-44: mind the subscript of CO2. 4. Line 43-44: I have not found the information of 30% in Behrenfield et al, 2005. 5. Line 48: The description "fish grazing on phytoplankton" is not accurate. The effect of phytoplankton on fisheries is via marine food chain, i.e. zooplankton grazing on phytoplankton provide food source for some fish. 6. Line 56: Please add the citation: Sosik, H.M.; Sathyendranath, S.; Uitz, J.; Bouman, H.; Nair, A. In situ methods of measuring phytoplankton functional types. In Phytoplankton Functional Types from Space. Reports of the International Ocean-Colour Coordinating Group (IOCCG), No. 15; Sathyendranath, S., Ed.; IOCCG: Dartmouth, NS, Canada, 2014; pp. 21–38. 7. Line 84: use the abbrevation of "PSC". Full name is not needed. 8. Line 86: the term "PSC percentage" is inaccurate. It is the contributions of Chla from different phytoplankton size classes to total Chla concentration. 9. Line 105: the colour of the land is not red. 10. Line 111: delete "a". 11. Line 112: "systems". 12. Line 161: "wavelengths". 13. Please define the abbreviation of a variable before using it (e.g. Table 1 and a lot of places). 14. Line 181-182: this is not a sentence. 15. Line 182: typo: divinyl chl-a. Did you consider chlorophyllide-a as part of Tchl-a? 16. Line 186-190: you have mentioned these in Line 113-117. 17. Figure 4&5: Rrs is not defined. From my perspective, extensive editing of English language and style required.

I hope you find my comments helpful for your revision. Best regards, Yangyang Liu Alfred Wegener Institute

---

## Author Comment (AC1) · 14 May 2019

We thank the reviewer for his helpful comments. We answer point by point: We use the following typo : The reviewer comments are in italic Our answers are in normal typo

SPECIFIC COMMENTS 1 Line 50-52 we rewrote these lines: Microscopy is time consuming and is unable to identify picoplankton. Imaging flow cytometry (IFC) has renewed microscopic methods, thanks to the speed at which they are able to characterize phytoplankton in a water sample (Sosik et al, IOCCG report n°15, 2014).

[Figure]

2-3-4-5 Pigments allow to estimate phytoplankton groups but not phytoplankton species. We withdrew this statement in the text. Lines -53-60 we rewrote this section: An alternative method is the analysis of seawater samples by high-performance liquid chromatography (HPLC) which is widely used to categorize broad phytoplankton groups such as PFT or PSC (Jeffreys et al, 1997, Brewin et al, 2010, Hirata et al, 2011). HPLC enables identification of 25 to 50 pigments within a single analysis, which is much easier and faster to conduct than microscopic observations. Each phytoplankton group is associated with specific diagnostic pigments and a conversion formula can be derived to estimate the percentage of each group from the pigment measurements (Vidussi et al, 2001; Uitz et al, 2010). HPLC measurements are now recognized as the standard for calibrating and validating satellite-derived chlorophyll-a concentration and for mapping groups of phytoplankton (Sosik et al, IOCCG report n°15, 2014).

2-Lines 139-140 , we added in the revised version of the manuscript Matchup procedure between in situ and satellite observation is a crucial question to estimate remote sensing algorithms. If the parameters of the procedure are too severe, the number of collocated data is dramatically decreasing. If the parameters are too large, the accuracy of the matching is decreasing. We then chose some compromise. Usually people use a matchup window of 3X3 pixels (Alvain et al, 2005) which corresponds to a distance somewhat less than 20km between the satellite pixel and in situ measurement since we deal with level 3 satellite observations whose pixel is of the order of 9X9km. This criterium refers to the typical length of ocean variability (Levy et al, 2012; Levy, 2003)

3-Lines 150-160 and Figure 3. In figure 3, we present the regression line between Chla- given by OC4V4 and in situ Chla. The data are given in mgm-3 but the scale in figure 3 are log scales. Please use more statistical metrics in addition to R-square and RMSE according to Brewin et al, 2015: Brewin et al (2015) give a large variety of statistical parameters due to the fact that they compare a large number of models whose performances are close together, which implies the use of several criteria to

separate them. In the present study, we only need to estimate the quality of our model, which can be done by standard statistical parameters as usual. Besides the software we used to do the cross validations only computes the RMSE and the R2 parameters. Concerning the pigment concentrations, the statistical tests were done in mgm-3. We included this information in the text.

4-Lines 288-289: you have said the same as Line 264-265. We insist on that point because it constitutes the original component of 2S-SOM.

4-Table 2: often these statistics are done on log(pigments) - given their distribution and expected errors. Our strategy is to compute the statistical parameters in the physical space as most statisticians do and as did Brewin et al (2015) in order to facilitate the interpretation. The concentration values are normalized during the learning procedure of the SOM.

5-Line 402: Unfortunately, it cannot be concluded that diatoms dominated because of high Fuco ratio and chl-a, without additional information on phytoplankton groups using e.g. microscopy. We do not have concomitant microscopy measurements. We strongly think that the bottom right region in the 2S-SOM correspond to diatoms since high fucoxanthin is associated with high chlorophyll concentration and low peridinin. It is seen in Figures 8, 10 and 11 that high fucoxanthin geographical regions are situated near the coast where diatoms were observed in previous studies (Farikou et al, 2015; Blasco et al, 1980 ) whilst high peridinin geographical region are situated in offshore regions. We changed our previous sentence in : Moreover, the bottom right region in the 2S-SOM may correspond to the diatoms with a good confidence since high fucoxanthin is associated with high chlorophyll concentration and low peridinin. This is endorsed in section 5 of the present paper by looking at the geographical location of the different pigment concentrations (figures 8, 10, 11).

6. Please spell MLP out in the Discussion section. MLP stands for Multi Layer Perceptron

7. Line 649-654: Can you summarize why SOM needs fewer data points than MLPs and other super- vised learning? Why MLP cannot be trained with total âĹij500 dapa points?

This is a well-known property of SOM versus MLP. The main difference between MLP and SOM is in the learning process: MLP is a supervised algorithm while SOM uses un-supervised learning. Both have to estimate a large number of weights during a learning phase; the accuracy of the methods depends on the dimension of the input and output spaces, the number of data available and the number of weights to estimate. In SOM the weights are highly regularized by the neighborhood function, so the number of data needed for learning is less that for the MLP. In the present application the MLP would have to approximate a highly non-linear function from the R11 input space (the remote sensing parameters) to the R6 output space that represents the pigments. Due to the highly non-linearity of the function, the 515 data available for the learning is too small to adequately sample the R11 space of the function. In the other hand, SOM is not a regressor but uses automatic clustering methods and provides more robust values. Moreover, the topological order prevents to make errors in interpolating between two clusters. We think this explanation is too long to be included in the present text and out of the scope of the present study. It would be relevant in a Text Book or a review paper dedicated to NN. We propose to escape this question and to withdraw the sentence at line 650 : 'which makes MLPs and classical supervised learning methods unusable' The sentence is now : 'We used an unsupervised neural network classification method which is an extension of the SOM method well adapted to deal with small database whose elements are very inhomogeneous'

8. Is it possible to clarify the minimum threshold of pigment concentration of the appli-cability of 2S-SOM? The minimum and maximum values of a parameter are those of the learning data base. As the SOM use has 162 neurons, the interval between the minimum and maximum values is divided in 162 discrete values corresponding to the values captured by the referent vector associated with each neuron. We get these discrete values empirically only by looking at the different referent vector of the SOM. The discrete values are computed during the learning phase and depend on the density of the learning data set: sparse regions are les sampled than dense ones.

TECHNICAL CORRECTIONS

1 We homogenized the spelling of Sénégal in the revised version 2, 3 OK, we did the correction 4 lines 43-44 We put as a reference for the rate of $CO_2$ captured by the OCEAN : Le Quéré et al, 2018 which is more appropriate 5 line 48 We changed the sentence as : and fisheries with a possible effect on fish grazing on phytoplankton via the marine food chain 6, 7 OK, we did the correction 8 line 86 We modified the sentence as : ' These algorithms try to establish a relationship between the chl-a concentration and the chl-a concentration fractions associated with each of the three PSC' 9 We changed 'red' into 'brown' 10, 11, 12, 13: OK, we did the correction suggested 14 We modified the line 181 : 'which is defined as the ratio of the diagnostic pigment (DP) versus the total chl-a'. 15 We used the definition of Alvain et al (2005), where Chl-a is part of Tchl-a (Tchl-a= Chl-a+ Divinyl chlorophyll-a). 16 We delete the sentence in lines 186-190 17 Rs stands for ïĄšw(ïĄň), we made the change in figures 4 and 5 in the revised version

Added references

Levy, M., Mesoscale variability of phytoplankton and of new production: Impact of the large-scale nutrient distribution, J. Geophys. Res., 108(C11), 3358, doi:10.1029/2002JC001577, 2003.

Lévy M. , D. Iovino, L. Resplandy, P. Klein, G. Madec, A.-M. Tréguier, S. Masson, K. Takahashi, (2012) Large-scale impacts of submesoscale dynamics on phytoplankton: Local and remote effects, Ocean Modelling,77–93

Le Quéré et al, (2018) : Global Carbon Budget 2018, Earth Syst. Sci. Data, 10, 2141–2194, 2018 ; https://doi.org/10.5194/essd-10-2141-2018

Sosik, H.M.; Sathyendranath, S.; Uitz, J.; Bouman, H.; Nair, A. (2014) In situ methods of measuring phytoplankton functional types. In Phytoplankton Functional Types from Space. IOCCG report, No. 15; Sathyendranath, S., Ed.; IOCCG: Dartmouth, NS, Canada, 2014; pp. 21–38.
* * *

---

## Referee Comment (RC2) · Anonymous Referee #2 · 3 Nov 2019

This paper presents a new neural classifier model for estimating phytoplankton pigments from satellite data, by using an unsupervised learning method instead of the standard supervised learning approach, as the sample size is limited. The method involves blocking the variables as opposed to the unblocked approach in the standard self-organizing map (SOM) approach. I believe the writing needs considerable improvement, as the paper is difficult to read, and the advantage of their 2S-SOM over the standard SOM is not brought out. The English can also be improved.

Specific comments:

[Figure]

There is a lack of comparison with controls for the reader to appreciate the advantage of using this new model. At the minimum, there should be more comparison between the new 2S-SOM model performance scores versus the standard SOM model scores. [The paper would be more interesting if the performance of 2S-SOM is also compared against standard supervised learning models such as multi-layer perceptrons or random forests.]

On line 321, the choice of the elongated 2-dimensional grid of 9x18 is not obvious. Why is a more square grid (e.g. 12x12, 12x13 or 13x13) not used?

The paper is very hard to read as there is a tendency to present many undefined symbols all at once, with the symbols remaining undefined until much later in the paper. For instance, Eq.(5) introduces a large number of symbols and terms all at once. The "block" is not explained in a concrete way until the next section (Sect. 3.3), so I had a misconception on how the data were blocked when reading Sect. 3.2. A much more logical order of presentation is to present the concept of blocking variables first, and try to explain as many of the symbols coming up in Eq.(5) before actually presenting the equation. Also around Eq.(5), there are numerous typos and inconsistent fonts (as listed later in this review).

Minor comments/typos:

Line 22-23: "Thanks to . . . new method. It primarily consists in. . ." is verbose. Simplify to "Our new method consists of . . ."

Line 25: "carried using" should be "carried out using".

Line 69 and throughout the manuscript: Bold fonts are for vectors and matrices (see the journal's manuscript preparation guidelines), but here they are often used for scalars and units. There are many places where the font switches back and forth between bold and Roman and italics (e.g. lines 248-251 and line 272).

Line 151: Need a reference for the OC4V4 algorithm.

Line 162: The last sentence of the paragraph and Table 1 need to be moved to after line 183. The table is currently placed before the terms in it are defined.

Line 174: How can Ra be independent of chl-a if it is divided by rho_wref which is dependent on chl-a?

Line 174: "sensitive the" should be "sensitive to the".

Line 248: W is undefined.

Line 254: should give a specific reference on the kernel and temperature.

Line 276: How were B and Pb chosen?

Line 278: "a" should be alpha.

Line 282: Eta should be beta.

Figure 4: For 2S-SOM, I can see long dash, short dash, space and no space variants.

Line 420: Last sentence of paragraph: I have trouble understanding this sentence.

Fig.13: Top right corner is slightly chopped off.

Line 542: "a" should be alpha.

Fig.16: I don't understand why the black curve tends to lie closer to the blue curve than the red curve is to the blue curve. I would have expected the red curve to lie closer to the blue curve. I might have misunderstood what the curves represent – please give more detailed explanation.

Line 639: Replace "people" with "studies".

---

## Author Comment (AC2) · 20 Dec 2019

Answers to reviewer n°1

We first thank the reviewer for his helpful comments and suggestions that have helped us to improve the manuscript. In the following, we answer point by point using the following convention: The reviewer comments are in italic Our answers are in standard typo The changes we made according to the recommendation of reviewer 1 of are in yellow in the track document.

[Figure]

1. The introduction can be written in a much more accurate way. I would check it phrase by phrase and sentence by sentence. We rewrote the introduction taking into account the remarks of the reviewer

1.1. Line 50-52 For example, one limitation of microscopy is the difficulty in indentifying picoplankton 1.2. The optical microscopy method is developing, for example the imaging flow cytometry (IFC). We rewrote these lines: "Microscopy is time-consuming and is unable to identify picoplankton. Imaging flow cytometry (IFC) has renewed microscopic methods, thanks to the speed at which they are able to characterize phytoplankton in a water sample (IOCCG report n°15, 2014)". (Lines 49-52 in the revised version).

1.3. Line 54-55: Mind the use of the terms PSC and PFT. PFT depends on how you define it. PSC is also a type of PFT definitions. Pigments allow estimating phytoplankton groups but not phytoplankton species. We withdrew this statement in the text.

1.4. Line 57-60: the conversion formula method is the so-called "Diagnostic Pigment Analysis". CHEMTAX uses matrix factorization to estimate PFT from pigments. We mentioned the so-called "Diagnostic Pigment Analysis" line 57 1.5. Line 60: I am not sure with just marker pigments themselves the identification of phytoplankton can be achieved in species level. We agree and we, therefore, modified the text of the revised version

1.6. In summary, please check IOCCG report 15 and related literature carefully. According to comments n°3, 4, 5, 6 we rewrote these lines which are now (Lines 52-61 in the revised version) taking into account the material in the IOCCG report 15: "An alternative method is the analysis of seawater samples by high-performance liquid chromatography (HPLC) which is widely used to categorize broad phytoplankton groups such as PFT or PSC (Jeffreys et al, 1997, Brewin et al, 2010, Hirata et al, 2011). HPLC enables identification of 25 to 50 pigments within a single analysis, which is much easier and faster to conduct than microscopic observations. Each phytoplankton group is associated with specific diagnostic pigments and a conversion formula can

be derived to estimate the percentage of each group from the pigment measurements (Vidussi et al, 2001; Uitz et al, 2010). HPLC measurements are now recognized as the standard for calibrating and validating satellite-derived chlorophyll-a concentration and for mapping groups of phytoplankton (IOCCG report n°15, 2014)".

2-

2. Lines 139-140 Match-up procedure can be more detailed, for example, by adding the criteria of refusing data points and the reason why you choose 20km We rewrote these lines in the revised version of the manuscript (lines 138-151) "Matchup procedure between in situ and satellite observation is a crucial question to estimate remote sensing algorithms. If the parameters of the procedure are too severe, the number of collocated data is dramatically decreasing. If the parameters are too large, the accuracy of the matching is decreasing. We then chose some compromise. Usually, people use a matchup window of 3X3 pixels (Alvain et al, 2005) which corresponds to a distance somewhat less than 20km between the satellite pixel and in situ measurement since we deal with level 3 satellite observations whose pixel is of the order of 9X9km. This criterium refers to the typical length of ocean variability (Levy et al, 2012; Levy, 2003)"

3. Lines 150-160 and Figure 3. Please use more statistical metrics in addition to R-square and RMSE according to Brewin et al 2015. Please specify whether they are calculated in log scale or not. Brewin, Robert JW, et al. "The Ocean Colour Climate Change Initiative: III. A round-robin comparison on in-water bio-optical algorithms." Remote Sensing of Environment 162 (2015): 271-294 Brewin et al (2015) give a large variety of statistical parameters because they compare a large number of models whose performances are close together, which implies the use of several criteria to separate them. In the present study, we only need to estimate the quality of our model, which can be done by standard statistical parameters as usual. Concerning the pigment concentrations, the statistical tests were done in mg.m-3. We included this information in the text (lines 181-183).

In figure 3, we present the regression line between Chla- given by OC4V4 and in situ chl-a. The data are given in mg.m-3 and the statistical estimators were computed in mg.m-3 but the scale in figure 3 is log scales.

4. Lines 288-289: you have said the same as Line 264-265. We insist on that point because it constitutes the original component of 2S-SOM.

4. Table 2: often these statistics are done on log(pigments) - given their distribution and expected errors. Our strategy is to compute the statistical parameters in the physical space as most statisticians do and as did Brewin et al (2015) to facilitate the interpretation. The concentration values are normalized during the learning procedure of the SOM.

5. Line 402: Unfortunately, it cannot be concluded that diatoms dominated because of high Fuco ratio and chl-a, without additional information on phytoplankton groups using e.g. microscopy. We do not have concomitant microscopy measurements. When analyzing the referent vectors presented in Fig 6, we strongly think that the bottom right region representing the neurons of the 2S-SOM may correspond to diatoms since high fucoxanthin is associated with high chlorophyll concentration and low peridinin. Besides, it is seen in Figures 8, 10 and 11 that high fucoxanthin geographical regions are situated near the coast where diatoms were observed in previous studies (Farikou et al., 2015; Blasco et al., 1980) while high peridinin geographical regions are situated in offshore regions. We changed our previous sentence in: 'Moreover, the bottom right region in the 2S-SOM may correspond to the diatoms with good confidence since high fucoxanthin is associated with high chlorophyll concentration and low peridinin. This is endorsed in section 5 by looking at the geographical location of the different pigment concentrations (figures 8, 10, 11)'. (Lines 352-356 of the revised version)

6. Please spell MLP out in the Discussion section. MLP stands for Multi LayerPerceptron, it has been added on line 596

7. Line 649-654: Can you summarize why SOM needs fewer data points than MLPs

and other supervised learning? Why MLP cannot be trained with a total of âĽij500 data points? This is a well-known property of SOM versus MLP. The main difference between MLP and SOM is in the learning process: MLP is a supervised algorithm while SOM uses unsupervised learning. Both have to estimate a large number of weights during a learning phase; the accuracy of the methods depends on the dimension of the input and output spaces, the number of data available and the number of weights to estimate. In SOM the weights are highly regularized by the neighborhood function, so the number of data needed for learning is less than for the MLP. In the present application, the MLP would have to approximate a highly non-linear function from the R11 input space (the remote sensing parameters) to the R6 output space that represents the pigments. Due to the highly non-linearity of the function, the 515 data available for the learning is too small to adequately sample the R11 space of the function. On the other hand, SOM is not a regressor but uses automatic clustering methods and provides more robust values. Moreover, the topological order prevents to make errors in interpolating between two clusters. We think this explanation is too long to be included in the present text and out of the scope of the present study. It would be relevant in a Text Book or a review paper dedicated to NN. We propose to escape this question and to withdraw the sentence at line 650: 'which makes MLPs and classical supervised learning methods unusable' The sentence is now: 'We used an unsupervised neural network classification method which is an extension of the SOM method well adapted to deal with a small database whose elements are very inhomogeneous'(lines 605-607 of the revised version)

8. Is it possible to clarify the minimum threshold of pigment concentration of the applicability of 2S-SOM? The minimum and maximum values of a parameter are those of the learning data base. As the 2S-SOM has 162 neurons, the interval between the minimum and maximum values is divided into 162 discrete values corresponding to the values captured by the referent vector associated with each neuron. Classification acts as a piecewise continuous model permitting the achievement of complex tasks. We get these discrete values empirically only by looking at the different referent vectors of the

SOM.

TECHNICAL CORRECTIONS

1. The country Senegal has three versions of names in the manuscript, i.e. SeÌ ËŻAneÌ ËŻAgalo (title), Senegalo (context) and Senegal (Figure 1). Please keep the consistency. We homogenized the spelling of Senegal in the revised version

2. line 41 The word "phytoplankton" is more often used as a plural modified (line40, 41, 49 of the revised version)

3. Line 42-44: mind the subscript of $CO_2$ modified

4. lines 43-44: I have not found the information of 30% in Behrenfield et al, 2005 We put a more appropriate reference for the rate of $CO_2$ captured by the ocean: "Le Quéré et al, 2018" (line 43)

5. line 48: The description "fish grazing on phytoplankton" is not accurate. The effect of phytoplankton on fisheries is via marine food chain, i.e. zooplankton grazing on phytoplankton provide food source for some fish. We changed the sentence as: "and fisheries with a possible effect on fish grazing on phytoplankton via the marine food chain" (line 46-47 of the revised version)

6. Line 56: Please add the citation: Sosik, H.M.; Sathyendranath, S.; Uitz, J.; Bouman, H.; Nair, A. In situ methods of measuring phytoplankton functional types. In Phytoplankton Functional Types from Space. Reports of the International Ocean-Colour Coordinating Group (IOCCG), No. 15; Sathyendranath, S., Ed.; IOCCG: Dartmouth, NS, Canada, 2014; pp. 21–38 Done (line 56 in the revised version)

7. Line 84: use the abbreviation of "PSC". Full name is not needed Done

8. line 86: the term "PSC percentage" is inaccurate. It is the contributions of Chla from different phytoplankton size classes to total Chla concentration We modified the sentence as: ' These algorithms try to establish a relationship between the chl-a con-
centration and the chl-a concentration fractions associated with each of the three PSC'
(lines 86-88 of the revised version)

9. Line 105: the colour of the land is not red. We changed 'red' into 'brown

10. Line 111: delete "a". 11. Line 112: "systems". 12. Line 161: "wavelengths". 13.
Please define the abbreviation of a variable before using it (e.g. Table 1 and a lot of
places). We implemented the suggested corrections.

14. lines 181-182: this not a sentence We modified this line which is now 'which is
defined as the ratio of the diagnostic pigment (DP) versus the total chl-a'.(lines 178-
179 of the revised version)

15. Line 182: typo: divinyl chl-a. Did you consider chlorophyllide-a as part of Tchl-a?
We used the definition of Alvain et al (2005), where Chl-a is part of Tchl-a (Tchl-a=
Chl-a+ Divinyl chlorophyll-a). (line 179)

16. Line 186-190: you have mentioned these in Line 113-117 We delete the sentence
in lines 186-190

17. Figure 4&5: Rrs is not defined. Rrs stands for ïĄšw(ïĄň), we made the change in
figures 4 and 5 in the revised version

The manuscript has been read and corrected by a native English-speaking person

Added references Levy, M., Mesoscale variability of phytoplankton and of new produc-
tion: Impact of the large-scale nutrient distribution, J. Geophys. Res., 108(C11), 3358,
doi:10.1029/2002JC001577, 2003.

M. Lévy, D. Iovino, L. Resplandy, P. Klein, G. Madec, A.-M. Tréguier, S. Masson, K.
Takahashi, (2012) Large-scale impacts of submesoscale dynamics on phytoplankton:
Local and remote effects, Ocean Modelling,77–93

Le Quéré et al, (2018) Global Carbon Budget 2018, Earth Syst. Sci. Data, 10, 2141–
2194, 2018 ; https://doi.org/10.5194/essd-10-2141-2018

Sosik, H.M.; Sathyendranath, S.; Uitz, J.; Bouman, H.; Nair, A. In situ methods of measuring phytoplankton functional types. In Phytoplankton Functional Types from Space. IOCCG report, No. 15; Sathyendranath, S., Ed.; IOCCG: Dartmouth, NS, Canada, pp. 21–38, 2014.
* * *

---

## Author Comment (AC3) · 20 Dec 2019

We first thank the reviewer for his helpful comments and suggestions that have helped us to improve the manuscript. In the following, we answer point by point using the following convention: The reviewer comments are in italic Our answers are in standard typo The changes we made according to the recommendation of reviewer 2 of are in turquoise in the track document

There is a lack of comparison with controls for the reader to appreciate the advantage

of using this new model. At the minimum, there should be more comparison between-scewise the new 2S-SOM model performance scores versus the standard SOM model scores. [The paper would be more interesting if the performance of 2S-SOM is also compared against standard supervised learning models such as multi-layer perceptrons or random forests.]

We comment on the advantages/disadvantages of the different methods in the discussion section (line 594-609). An objective comparison of the different methods is out of the scope of the present paper as it would considerably increase the length of the present paper. In fact, it would deserve a full paper (see the paper of Brewin et all (2011) dedicated to a comparison of the different methods and also the paper of Bracher et al, 2017, Obtaining Phytoplankton Diversity from Ocean Color: A Scientific Roadmap for Future Development. Front. Mar. Sci. 4:55.). Besides to be conclusive, such a comparison should be done on a specific region where in situ measurements are more numerous than in the present region. We first used a SOM and then decided to use a 2S-SOM mainly by the information provided by the 2S-SOM on the role of the different variables in the classification process. The major advantage of the 2S-SOM compared with the SOM and other classification methods is to partition the different variables of the dataset under study into blocks and to affect weights to these blocks. The block weighting facilitates the clustering procedure by favoring the taking into account of the most pertinent variables. This method is related to the research area developed in statistics under the designation of clusterwise method (Parson et all 2004; Kriegel et all 2009)

Parsons L, Haque E et Liu H : Subspace clustering for high dimensional data : a review. SIGKDD Explor. Newsl., pages 90105, 2004. ISSN 1931-0145. 73, 74, 80 Kriegel H.-P, Kröger P et Zimek A : Clustering high-dimensional data : A survey on subspace clustering, pattern-based clustering, and correlation clustering. ACM Trans. Knowl. Discov. Data, 3(1):1 :11 :58, mars 2009. ISSN 1556-4681. 37, 73, 74, 80

A high weight affected to a block means that the associated variables play a major role

in the classification process; a small value means that the associated variable plays a minor role: this information is of importance to identify the variables which control the process under study. Besides the block weighting provides useful information on the functioning of the classification by permitting to identify the variables which control it and allows us to better understand the dynamics of the phytoplankton communities. This is discussed in lines 371-376 of section 4-2 Analysis of the topology of the 2S-SOM corresponding to the analysis of figure 7 showing the different weights affected to the neurons of the 2S-SOM also in lines 494-509 of section 5 corresponding to the analysis of figure 13, and in line 622-627 of the discussion section. Moreover, we added the block weights $\alpha$ as an output of 2S-SOM in figures 4 and 5

On line 321, the choice of the elongated 2-dimensional grid of 9x18 is not obvious. Why is a more square grid (e.g. 12x12, 12x13 or 13x13) not used?

The size of the map has been determined (line 275-276, added in the new version), by using the SOM software http://www.cis.hut.fi/projects/somtoolbox/download/, assuming that the size of SOM and 2S-SOM depend on the same criteria. We also checked other grid configuration and found that the most efficient is the 9x18 neurons

The paper is very hard to read as there is a tendency to present many undefined symbols all at once, with the symbols remaining undefined until much later in the paper. For instance, Eq.(5) introduces a large number of symbols and terms all at once. The "block" is not explained in a concrete way until the next section (Sect. 3.3), so I had a misconception on how the data were blocked when reading Sect. 3.2. A much more logical order of presentation is to present the concept of blocking variables first, and try to explain as many of the symbols coming up in Eq.(5) before actually presenting the equation. Also around Eq.(5), there are numerous typos and inconsistent fonts (as listed later in this review).

We have rewritten the sections 3.1 and 3.2 describing the functioning of the SOM and 2S-SOM. We put in Annex the mathematical description of that functioning. In the main

text, we only describe the principle of the functioning of the 2S-SOM. We now explain all the symbols we used. The blocks are described in the main text (lines 255-260) before the explanation of the functioning of the SOM and the 2S-SOM, which is in Annex. We also focused attention on the typos

Line 22-23 Thanks to . . . new method. It primarily consists in. . ." is verbose. Simplify to "Our new method consists of .

done (line 22)

Line 25 "carried using" should be "carried out using".

done (line 24)

Line 69 and throughout the manuscript: Bold fonts are for vectors and matrices (see the journal's manuscript preparation guidelines), but here they are often used for scalars and units. There are many places where the font switches back and forth between bold and Roman and italics (e.g. lines 248-251 and line 272).

We carefully read the manuscript and corrected the font errors

Line 151: Need a reference for the OC4V4 algorithm.

We give a reference for the OC4V4 algorithm (O'Reilly et al, 2001) – (line 154 in the revised version)

Line 162 The last sentence of the paragraph and Table 1 need to be moved to after line 183. The table is currently placed before the terms in it are defined.

Done

Line 174: How can Ra be independent of chl-a if it is divided by rho_wref which is dependent on chl-a?

Ra, which is defined as rhow(lambda)/rhow wref(Lambda, chl-a) is the key parameter of the Physat method (Alvain et al, 2005, 2012). rhow(lambda)depends on secondary phytoplankton pigments + chl-a, while )/rhow wref(Lambda, chl-a) depends on chl-a only. The reasoning of Alvain et al (2005) is that the ratio rhow(lambda)/rhow wref(Lambda, chl-a) depends on secondary phytoplankton only since both depend on chl-a.

Line 248: W is undefined.

W is now defined in line 230

Line 254: should give a specific reference on the kernel and temperature.

References are given in lines 930, 931 of Annex (Kohonen, 2001, Niang et al, 2003)

Line 276: How were B and Pb chosen?

These variables are defined in lines 941 and 944: B is the number of blocks (B=4) and Pb is the number of variables in block b. According to the definition of blocks (lines 257-262), P1= 5, P2= 5, P3= 5, P4= 2.

Line 278: "a" should be alpha.

Corrected (line 953 in the new version)

Line 282: Eta should be beta.

Corrected (line 953 in the new version)

Figure 4: For 2S-SOM, I can see long dash, short dash, space and no space variants.

Figure 4. We check the pdf output corresponding the figure 4. It seems ok in the modified version. Perhaps there was a software problem in the conversion of the original text written in Word into pdf.

Line 420: Last sentence of paragraph: I have trouble understanding this sentence.

We changed this sentence and gave more explanation on the description of the 2S-SOM neurons. The sentence is now (Line 381-384 in the new version): "These neurons correspond to very small chl-a concentrations, which are estimated with large errors. Besides, we remark that high ïĄą values for chl-a correspond to high chl-a concentration values (bottom right of the chl-a panel in figure 7 and figure 6 respectively). For these cases, the clustering assembled data that mainly depend on chl-a concentration".

Fig.13: Top right corner is slightly chopped off.

Done.

Line 542: "a" should be alpha.

Done. We replaced 'a weight' by 'a weight ïĄą' which is clearer (line 497 of the new version)

Fig.16: I don't understand why the black curve tends to lie closer to the blue curve than the red curve is to the blue curve. I would have expected the red curve to lie closer to the blue curve. I might have misunderstood what the curves represent – please give more detailed explanation.

A VIIRS sensor observation is captured by a neuron of the 2S-SOM whose associated referent spectrum is the red curve in figure 16. This referent spectrum is the mean of the different spectra captured by that neuron during the learning phase. Among these different spectra, there is one (black curve in figure 16) which is the closest to the UPSEN spectrum (blue curve in figure 16). It is expected that the black curve is closer to the blue curve than the red curve which is flattened due to the averaging process. We reformulated this description in the text which was not clear in the first version. (line 539-546).

Line 639: Replace "people" with "studies".

Done (line 596 of the revised version).
* * *

---

## Referee Report (RR1)

Specific comments:

Line 41-43:

The reference Le Quéré et al, 2018 does not provide this information. The oceans presently take up about 26% of the anthropogenic CO2, and phythonplankton contribute to this via the biological pump. However, the physical pump also plays an important role in the ocean-atmosphere CO2 exchange. The role of phytoplanton on regulating the climate should be described in a much more precise way.

Line 49-50:

1) "microscopy" should be revised as "light microscopy". Microscopy for identifying phytoplankton includes light, electron and epifluoroscence microscopy. Epifluoroscence microscopy and electron microscopy enable the identification of picophytoplankton.
2) Replace "unable" with "very difficult".
3) It is not appropriate to say IFC "renewed" microscopy. Faster samplying speed is an advantange of IFC over microscopy. However, both methods have their own limitations.

More information can be found in IOCCG report 2014 and Nair et al 2008.
Nair, A., et al., (2008). Remote sensing of phytoplankton functional types. Remote Sensing of Environment, 112(8), 3366-3375.

Line 53-54, 84, 91:

full names and abbreviations of PFT and PSC.

Line 58-59:

References Vidussi etal, 2001 and Uitz et al, 2010 are 2 examples of DPA.

Line 60:

IOCCG report 2014 does not provide the information that HPLC measurements are recognized as STANDARD for mapping phytoplankton groups. It originally says "HPLC measurements are now recognized as the standard for calibrating and validating satellite-derived chlorophyll-a concentration. Pigment analysis also has a useful role to play in validating satellite algorithms for mapping functional types of phytoplankton."

Line 48-61:

As far as I understand, this paragraph was meant to address the progress of methods for identifying phytoplankton, as mentioned in Line 48. However, instead of providing a relatively comprehensive review of the methods, the current manuscript only picked up 3 methods (i.e. light microscopy, IFC, HPLC) as the whole story "during the last two decades". I

would refer to the IOCCG report 2014 to the readers for the details and briefly summarize them in this paragraph.

Line 63-68:
It does not hurt if these lines are omitted.

Line 65-66:
light "transmitted" to satellite? "reflected back"?

Line 67:
1) water and non-algal particles also influence backsacttering and absorption.
2) add "and" before the last comma.

Line 68&76:
full name and abbreviation of "CDOM".

Line 69:
"This upwelling radiation" is confusing. Remote sensing reflectance is a ratio. Also, it is more commonly symbolized as "Rrs". "rho_w" more commonly stands for "water-leaving reflectance".

Line 76-78:
1) replace "detrital" to "non-algal".
2) b_b includes the contribution of water and particles (not just phytoplankton).
3) please describe exactly what G is related to.

Line96:
Please spell out 2S-SOM or at least SOM.

Line 99:
1) replace "sensed" to "sensing".
2) add "in situ" before "measurements".

Figure 1:
1) UPSEN campaign is not described (also in Line 122).
2) typo in the legend.
3) uppper right edge is cutted.

Section 2:
The method of measuring pigments is missing. The full names (Line 255-256) and abbreviations (Table 1) of pigments should also be mentioned.

Line 132-133 & 134-135:
repetitive.

Line 136, 140:
I may be wrong but I do not understand the word "severe".

Line 139, 151:
I have difficulty in understanding these sentences.

Line 147:
What do you mean by "parameters too large"? Does it mean the matchup window is too large?

Line 148:
1) replace "people" with "studies".
2) specify the spatial resolution of Seawifs, i.e. how big one pixel is.

Figure 3:
1) coefficient of vraisemblance?
2) delete one "in" at the end.

Line 164:
1) replace "having" with "with".
2) rho_w and R_a should not be called as "ocean reflectance".

Line 165:
add "respectively" for rho_w and R_a.

Line 167:
missed half bracket.

Line 170:
Please briefly describe how rho_wref is computed.

Line 191-192:
five rho_w and five R_a at five wavelengths?

Line 237-238:
I have difficulty in understanding these sentences.

Line 240:
Delete "frequently".

Line 243&282:
Please use the "distance" term consistently.

Section 3-2:
1) The 2S-SOM algorithm is the central part of the paper, and should then be addressed in detail directly in this section rather than in the appendix. The step-by-step procedures of applying this method to the dataset can be put in the appendix.
2) the currently description of the 2S-SOM method is not easy for me to understand, e.g would you possbily describe clearly how the dimensionality is reduced and how the weights are estimated? Also, adding an architecture of the 2S-SOM map would help.

Line 274:
How many classes in the case of 9x18 grids?

Line 300:
Again, the readers should not be sent to the appendix to check what mu and eta stand for.

Line 322:
How are the p-values of cross validation calculated?

Line 696:
concentrations.

Line 938:
k-means

Section 3-4.1:
please check again language and grammer mistakes.

I hope you'll find the comments helpful.

Best Regards,
Yangyang Liu (yangyang.liu@awi.de)